# Associative plasticity of granule cell inputs to cerebellar Purkinje cells

Rossella Conti, Céline Auger*

Université Paris Cité, CNRS, Saints-Pères Paris Institute for the Neurosciences, Paris, France

## eLife assessment

This study presents **valuable** findings on an unresolved question of cerebellar physiology: Do synapses between Purkinje cells and granule cells, made by the ascending part of the granule cells' axon, have different properties than those made by parallel fibers? The authors conducted patch-clamp recordings on rat cerebellar slices and found a new type of plasticity in the synapses of the ascending part of granule cell axons. The experiments are well-designed with appropriate controls, and the study provides **solid** evidence for the new form of cerebellar synaptic plasticity.

*For correspondence:
celine.auger@cnrs.fr

Competing interest: The authors declare that no competing interests exist.

**Abstract** Granule cells of the cerebellum make up to 175,000 excitatory synapses on a single Purkinje cell, encoding the wide variety of information from the mossy fibre inputs into the cerebellar cortex. The granule cell axon is made of an ascending portion and a long parallel fibre extending at right angles, an architecture suggesting that synapses formed by the two segments of the axon could encode different information. There are controversial indications that ascending axon (AA) and parallel fibre (PF) synapse properties and modalities of plasticity are different. We tested the hypothesis that AA and PF synapses encode different information, and that the association of these distinct inputs to Purkinje cells might be relevant to the circuit and trigger plasticity, similar to the coincident activation of PF and climbing fibre inputs. Here, by recording synaptic currents in Purkinje cells from either proximal or distal granule cells (mostly AA and PF synapses, respectively), we describe a new form of associative plasticity between these two distinct granule cell inputs. We show for the first time that synchronous AA and PF repetitive train stimulation, with inhibition intact, triggers long-term potentiation (LTP) at AA synapses specifically. Furthermore, the timing of the presentation of the two inputs controls the outcome of plasticity and induction requires NMDAR and mGluR1 activation. The long length of the PFs allows us to preferentially activate the two inputs independently, and despite a lack of morphological reconstruction of the connections, these observations reinforce the suggestion that AA and PF synapses have different coding capabilities and plasticity that is associative, enabling effective association of information transmitted via granule cells.

## Introduction

The ability to discriminate self-generated from external stimuli is essential to apprehend the world as movement itself activates sensory receptors. In fish, discrimination of self from external stimuli has been shown to rely on associating motor command and sensory feedback in cerebellar-like structures (*Montgomery and Bodznick, 1994*; *Sawtell, 2017*). This is achieved by modifying the motor inputs to generate a negative image of the associated sensory inputs, effectively subtracting the predictable sensory inputs. This way, only unpredicted inputs are output by the principal cells. While in most cases the role of the cerebellum in sensory information processing is still unclear, theoretical models and work on plasticity suggest that it might similarly be involved in cancellation of predictable sensory

inputs, but distinct sensory and motor synapses have yet to be identified in the cerebellum. The role of the cerebellum in motor learning and coordination has been well documented, but it is now clear that it is also involved in a number of higher cognitive functions (*Schmahmann and Sherman, 1998*; *Stoodley and Schmahmann, 2010*). An intriguing possibility, in the wider context of the cerebellum, is that receptive field and contextual inputs are encoded differentially by the granule cell AA and PF synapses. These inputs might be functionally associated based on relative timing to subtract predictable inputs, and output only unpredicted inputs, as observed for motor and sensory inputs in cerebellar-like structures.

Sensorimotor information is delivered to the cerebellar cortex by mossy fibres and relayed by granule cells (GCs) to the Purkinje cells (PCs) and inhibitory network for integration. Anatomically, the axon of a GC first extends through the PC layer and into the molecular layer, making up the ascending portion of the axon (AA), then bifurcates at a right angle, forming a PF that extends several millimetres along the folium in both directions. As a consequence, an AA can form multiple synapses with a few overlaying PCs in the same sagittal plane as the GC soma, distributing information to a restricted number of cells, whereas PFs course through the dendritic trees of hundreds of PCs along the lobule (*Napper and Harvey, 1988*), distributing information widely to other microzones. This morphology and functional data have suggested that PC activity might be principally driven by local GCs via AA synapses (*Cohen and Yarom, 1998*), while distant GCs might modulate its activity via the PF system to provide the PC with less specific contextual information.

In the adult rat, an estimated 175 k excitatory PF synapses encode information on the dendritic tree of a single PC (*Napper and Harvey, 1988*). We previously identified two PF synaptic populations with different molecular signatures (*Devi et al., 2016*). However, PF synapses have usually been treated as a uniform population, and we know little about the way the variety of information received by PCs is encoded at the synaptic level. *Isope and Barbour, 2002* estimated that 85% of PF synapses are silent, as suggested by in vivo data (*Ekerot and Jörntell, 2001*), however, that proportion was significantly lower for local GCs and presumably AA synapses. In this study and that by Walter (*Walter et al., 2009*), the properties of AA synapses were found to be indistinguishable from PF synapses. On the other hand, a study by *Sims and Hartell, 2005*, has shown that AA and PF synaptic properties and susceptibility to plasticity are different. AA synapses were shown to be refractory to plasticity with well-described protocols. Neither associative CF-mediated long term depression (LTD) nor LTP could be induced (*Sims and Hartell, 2005*; *Sims and Hartell, 2006*). These observations, together with the anatomical properties of the GC axon, suggest that AA and PF synapses can transmit different types of information and play different roles in cerebellar computation.

In this study, we asked whether coincident activation of synapses formed by proximal and distal GCs (mostly AA and PF synapses, respectively) might trigger specific synaptic plasticity of GC inputs. We show that a protocol pairing stimulation of AAs and PFs, with inhibition preserved, results in LTP of the AA inputs, while PF inputs are depressed although not significantly. AA-LTP is timing-dependent and relies on NMDAR and mGluR1 activation, bringing together these two pathways. GABA$_A$R activation was also required for efficient induction and maintenance of plasticity. Finally, we show that the different plasticity found for AA and PF inputs is not merely due to the differential spatial distribution of their synapses on the dendritic tree.

## Results

### Associative plasticity of AA synapses

To test for associative plasticity between ascending axon and parallel fibre inputs we positioned two separate stimulating electrodes to achieve activation of different GC inputs onto a recorded PC (*Figure 1A* top panel). To stimulate PFs, a stimulation pipette was positioned in the molecular layer, 100–200 μm away from the dendritic tree of the recorded cell. The average length of a PF is between 4.2 and 4.7 mm (*Pichitpornchai et al., 1994*). If we assume that only granule cells located within the 50 μm thick layer of the PC dendritic tree can make AA synapses, and take the average length of a parallel fibre, they represent only 1 to 2% of granule cells or parallel fibres. Therefore, the vast majority of fibres stimulated locally in the molecular layer belong to GCs distant from the recorded cell and only 1 to 2% are likely to originate from that 50 μm layer and make AA synapses. To stimulate AAs, a second stimulation pipette was positioned in the GC layer, within a narrow window centred on the

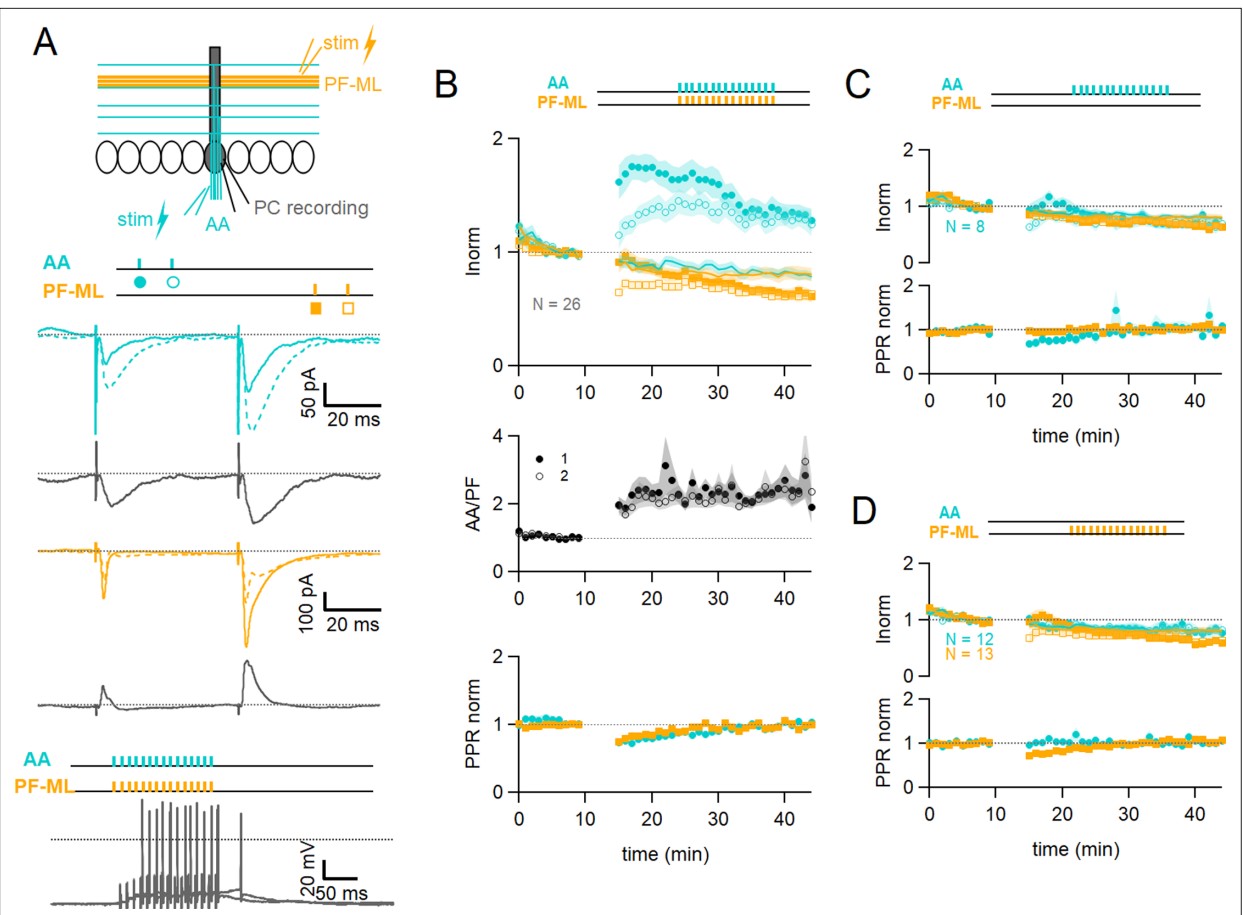

**Figure 1.** Associative plasticity of ascending axon (AA) and parallel fibre (PF) inputs. (**A, top**) Whole-cell recording in a Purkinje cell (PC). Two stimulation electrodes are used to activate granule cell (GC) inputs: one in the molecular layer to stimulate PFs (orange), and one in the GC layer to stimulate AAs (blue). The test stimulation protocol is depicted with colours and symbols as used in the plots in the following sections for the first (closed) and the second excitatory postsynaptic currents (EPSCs) (open). (**Middle**) AA- and PF-PSCs were sampled with a pair of pulses (dt=50 ms), every 10 s. Traces from one experiment: average AA-PSC (blue) and PF-PSC (orange), before (5–10 min, continuous line) and after the induction protocol (25–30 min, dotted line). Subtraction in grey. AA-EPSC amplitude increased while PF-EPSC amplitude decreased. No antagonist was applied. Evoked responses consisted of an EPSC often followed by an IPSC. (**Bottom**) induction protocol. Recording switched to current clamp, $V_H$=–65 mV. AAs and PFs are stimulated synchronously by a train of 15 pulses at 100 Hz every 3 s for 5 min. Grey traces, responses to the first two trains of stimulation. (**B Top**) a plot of the average AA- and PF-EPSC amplitude normalised to baseline for synchronous stimulation (5–10 min, n=26, colours and symbols as in A) and control No Stim experiments (continuous lines). Following induction, a long-term change in the inputs was observed. The amplitude of the AA-EPSC increased to 131 ± 7% (n=24) of baseline 25–30 min after induction (p=7e-5 one-tailed T-test, n=24). The PF-EPSC on the other hand decreased slowly to 65 ± 5% (n=25) of baseline (p=1e-7 one-tailed T-test, n=25). Continuous lines show the average time course of AA- and PF-EPSC amplitudes during control No Stim experiments where no stimulation was performed during the Iclamp period (n=17, see *Figure 1—figure supplement 1*), showing the extent of EPSC rundown during the course of the recordings. (**Middle**) the average ratio of the normalised amplitudes of AA- and PF-EPSCs (AA/PF), highlighting the relative change of the inputs, doubles. (**Bottom**) the average normalised paired-pulse ratio (AA2/AA1 and PF2/PF1) is transiently decreased following induction. (**C**) Average of eight experiments with stimulation of the AA pathway only during induction (labels, colours, and symbols as in A), and No Stim experiments overlaid (continuous lines). (**Top**) the normalised amplitudes of AA- and PF-EPSCs progressively decreased to 72 ± 7% (n=8) and 64 ± 6% (n=8) of baseline respectively, not significantly different from No Stim experiments (p=0.49 and p=0.48, respectively). Stimulation of the AA pathway alone is not sufficient to trigger AA-LTP. (**D**) Average of 13 experiments with stimulation of the PF pathway only during induction (labels), and No Stim experiments overlaid (continuous lines). (**Top**) the normalised AA-EPSCs showed a small and steady depression (84 ± 7% of baseline after 25–30 min) whereas the PF-EPSC depressed over time (64 ± 6%), not significantly different from No Stim experiments (p=0.63 and p=0.25, respectively). Stimulation of the PF pathway on its own is not sufficient to trigger plasticity. (**C** and **D bottom**) the PPR of the AA and PF pathways transiently decreased only for the pathway stimulated during induction. Values given are mean ± SEM. SEM is represented by shading. Statistical significance was tested using the Wilcoxon Mann Whitney test except for control data tested using T-test.

The online version of this article includes the following figure supplement(s) for figure 1:

**Figure supplement 1.** Control No Stim experiments.

**Figure supplement 2.** Effect of excitatory postsynaptic current (EPSC) amplitude and spiking of the Purkinje cell on plasticity.

PC somatodendritic plane. The GCs stimulated in this position have a high probability of making AA synapses onto the dendritic tree of the PC, although a small proportion could also make synapses from the proximal portion of PFs after bifurcation. For the sake of simplicity, we will use the terms AA and PF synapses thereafter although the terms proximal and distal synapses (with respect to GC soma) might be more appropriate.

The EPSC amplitude of AA and PF pathways were sampled with a pair of stimulations each (*Figure 1A* middle panel). On average, the amplitude of the AA-EPSC during baseline recordings was 110±12 pA (median 90 pA) and PF-EPSC 480 ± 100 pA (median 288 pA, n=25), and the paired-pulse ratio (PPR) was $PPR_{AA}$ 1.93 ± 0.08 and $PPR_{PF}$ 1.89 ± 0.07. Inhibitory inputs were preserved, and EPSCs were often followed by outward IPSCs. After baseline recording, a protocol aimed at inducing plasticity was applied (synchronous stimulation of both inputs by a train of 15 pulses at 100 Hz, every 3 s, 100 times, in the Current clamp, see *Figure 1A* bottom panel). Test of the AA and PF amplitude was then resumed. *Figure 1A* middle panel shows a sample experiment with average traces of the AA-PSC, in blue, and the PF-PSC, in orange, 5 min before (continuous line) the plasticity induction protocol, and 25–30 min after the protocol (dotted line). Subtraction traces (25–30 min - last 5 min) are shown in grey to highlight changes. The AA-EPSC amplitude is increased while the PF-EPSC amplitude is decreased. *Figure 1B* top panel shows the time course of 25 such experiments for the AA pathway and 26 for the PF pathway with amplitude normalised to the last 5 min of control before plasticity induction. It shows that on average, following the induction protocol, we observed a long-term change of opposite sign of the two inputs. The AA peak EPSC amplitude increased to 162 ± 15% of baseline immediately after the induction protocol and then slowly decayed within 15 min to stabilise at a value of 131 ± 7% 25–30 min after induction, significantly larger than the baseline (p=7e-5 one-tailed T-test, n=24). The PF-EPSC on the other hand was almost unchanged immediately after the protocol (91 ± 9%) and decreased to 65 ± 5% of baseline at 25–30 min, significantly smaller than the baseline (p=1e-7 one-tailed T-test, n=25).

*Figure 1B* also reveals a rundown of the EPSC amplitude during the baseline period, potentially obscuring long-term changes induced by plasticity. A series of control experiments was conducted to quantify the rundown (No Stim experiments, see *Figure 1—figure supplement 1*). After the baseline period, the recording was switched to Iclamp for 5 min, but no stimulation was delivered during that period. Continuous lines *Figure 1B* top panel shows the average time course for 17 No Stim experiments for the AA- (blue) and PF-EPSCs (orange). On average, 25–30 min after the Iclamp period, the AA-EPSC was 80 ± 6% of baseline (n=16, significantly smaller than baseline p=0.0004, and significantly smaller than synchronous stimulation protocol p=10e-7) and the PF-EPSC was 81 ± 8% of baseline (n=17, not significantly smaller than baseline p=0.062, and not significantly larger than synchronous stimulation protocol p=0.106), indicating a rundown of the amplitude for both pathways during experimental time, independent of the induction protocol. The extent of AA- and PF-EPSC rundown was not significantly different (n=16 and 17, respectively, p=0.82). These values can be used to correct long-term changes. When compensating for rundown, the plasticity protocol caused a long-lasting effective increase of the AA-EPSC to 164% of baseline and a decrease of the PF-EPSC to 80% of baseline on average.

*Figure 1B* middle panel shows the ratio of normalised AA and PF peak amplitudes (AA/PF). This measure highlights the relative amplitude change of AA and PF inputs, or relative plasticity, and eliminates changes common to both pathways, such as the rundown of synaptic responses. On average the normalised amplitude of the AA-EPSC more than doubles compared to that of the PF-EPSC, due to both the increase of the AA peak amplitude and the concomitant decrease of the PF, and this is sustained (AA/PF = 2.5 ± 0.4 with n=24, p=3e-4 after 25–30 min of induction). (*Figure 1B*) bottom panel shows the paired-pulse ratio of normalised amplitudes (PPR) for both pathways. While the peak amplitude of the first response in the pair of stimulations showed a progressive decrease, the peak amplitude of the second response of both AA and PF underwent either potentiation or depression respectively, and was relatively stable thereafter. As a result, the normalised PPRs are transiently decreased following induction, but return to their original value within 25–30 min ($PPR_{AA}$ 1.03 ± 0.03, $PPR_{PF}$ 0.99 ± 0.03, n=24, p=0.81). Changes in the PPR have been linked to a change in the presynaptic release probability (*Zucker and Regehr, 2002*). This suggests a transient presynaptic effect on release probability following induction and likely explains the progressive effects on AA and PF responses, but there was no significant long-term

change in the PPR, suggesting that the long-term effects observed are linked to postsynaptic changes.

Additionally, the level of plasticity measured in a given input pathway did not depend on the amplitude of the synaptic response of that input nor on the sum of the amplitudes of the two pathways (see *Figure 1—figure supplement 2B*), showing that even relatively small inputs can trigger plasticity. We analysed the average number of spikes and the time to the first spike during the first 5 trains of stimulation of the induction protocol. All cells except one did spike, even those with relatively weak synaptic currents for the first evoked response. The plastic change for the AA-EPSC only slightly correlated with the number of spikes (Pr=0.48) and the time to the first spike (Pr=0.4), and no correlation was observed for PFs (see *Figure 1—figure supplement 2C*).

To test whether the plasticity observed is due to the co-activation of AA and PF inputs or the specific properties of the stimulated inputs, we performed a series of experiments in which only one of the pathways was stimulated during the induction protocol. *Figure 1C* depicts the average behaviour in 8 experiments where only the AA pathway was stimulated during induction. The average amplitude of the AA-EPSC progressively decreased to 72 ± 7% (n=8) of control 25–30 min after the protocol, similar to the PF-EPSC (64 ± 6% of control, n=8), not different from No Stim experiments (p=0.49 and p=0.48, respectively). The normalised amplitude of the AA-EPSC was not significantly different from that of the PF-EPSC following induction (AA/PF: 1.2 ± 0.2 after 25–30 min, p=0.58, n=8, *Figure 1C* bottom panel). This shows that stimulation of the AA pathway alone is not sufficient to trigger AA-LTP. *Figure 1D* shows the average time course of 13 experiments where only the PF pathway was stimulated during induction. In this case, the AA pathway showed a decrease (84 ± 7% of baseline after 25–30 min, n=12), whereas the PF pathway developed a larger depression over time (64 ± 6% of baseline), but not different from No Stim experiments (p=0.63 and p=0.25, respectively). The relative amplitude of the normalised AA-EPSC (AA/PF) increased slowly after induction and was significant after 25–30 min (AA/PF=140 ± 11% of baseline, n=12, p=0.01), in line with the progressive depression of the PF pathway. In these experiments, the PPR of the AA and PF pathways were transiently modified after induction, as in control, but only for the pathway stimulated during the protocol (see *Figure 1C and D* bottom panel).

Stimulation of either pathway independently did not induce AA-LTP, showing that co-activation of AA and PF inputs is necessary, and AA-LTP is associative. The PF-EPSC decreased following the induction protocol but this was not significantly different from control No Stim experiments.

## Timing dependence of plasticity

The relative timing of inputs might be relevant to functional association, as observed for CF-mediated LTD (*Safo and Regehr, 2008*). We tested whether the relative timing of stimulation of the AA and PF pathways affects plasticity. During the induction protocol, the AA input was stimulated 150 ms after or 150 ms before the PF input. *Figure 2A* shows that when stimulating AAs 150 ms after PFs, the amplitude of the AA- and PF-EPSCs was depressed to a similar extent (AA was 53 ± 15% and PF 50 ± 20% of baseline, n=8, not significantly different from each other p=0.84), transforming AA-LTP into LTD. These changes were significantly different from experiments with synchronous stimulation (p=8e-5 for AA and p=0.022 for PF). The relative amplitude of the AA to PF pathway increased but not significantly (AA/PF was 224 ± 75% of baseline, n=8, p=0.2). On the other hand, when stimulating AAs 150 ms before PFs (see *Figure 2B*), the amplitude of the AA-EPSC was first facilitated and declined back to baseline (AA after 25–30 min was 97 ± 26%, n=7, significantly smaller than synchronous stimulation p=0.011), and the amplitude of the PF-EPSC was now close to baseline (84 ± 12%, n=7, not significantly larger than synchronous stimulation, p=0.062). The relative amplitude of the AA to PF pathway was not facilitated significantly (AA/PF was 126 ± 31% of baseline, p=0.35), while the PPR slightly increased (PPR$_{AA}$=114 ± 6%, p=0.008, and PPR$_{PF}$=107 ± 5%, n=7 and p=0.09, respectively).

In summary (*Figure 2C*), following the induction protocol the AA-EPSC was 131 ± 7% (n=24) of the baseline when AA and PF stimulation was synchronous; it was 53 ± 15% when AA stimulation was given 150 ms after PF stimulation (n=8), and 97 ± 26% (n=7) when the PF stimulation was started 150 ms after AA stimulation. The PF-EPSC was 65 ± 5% of baseline (n=25, significantly different from AA, p=6e$^{-11}$) when stimulation was synchronous, 50 ± 20% when AA stimulation was delayed by 150 ms (n=8), and 84 ± 12% when PF stimulation was delayed by 150 ms (n=7).

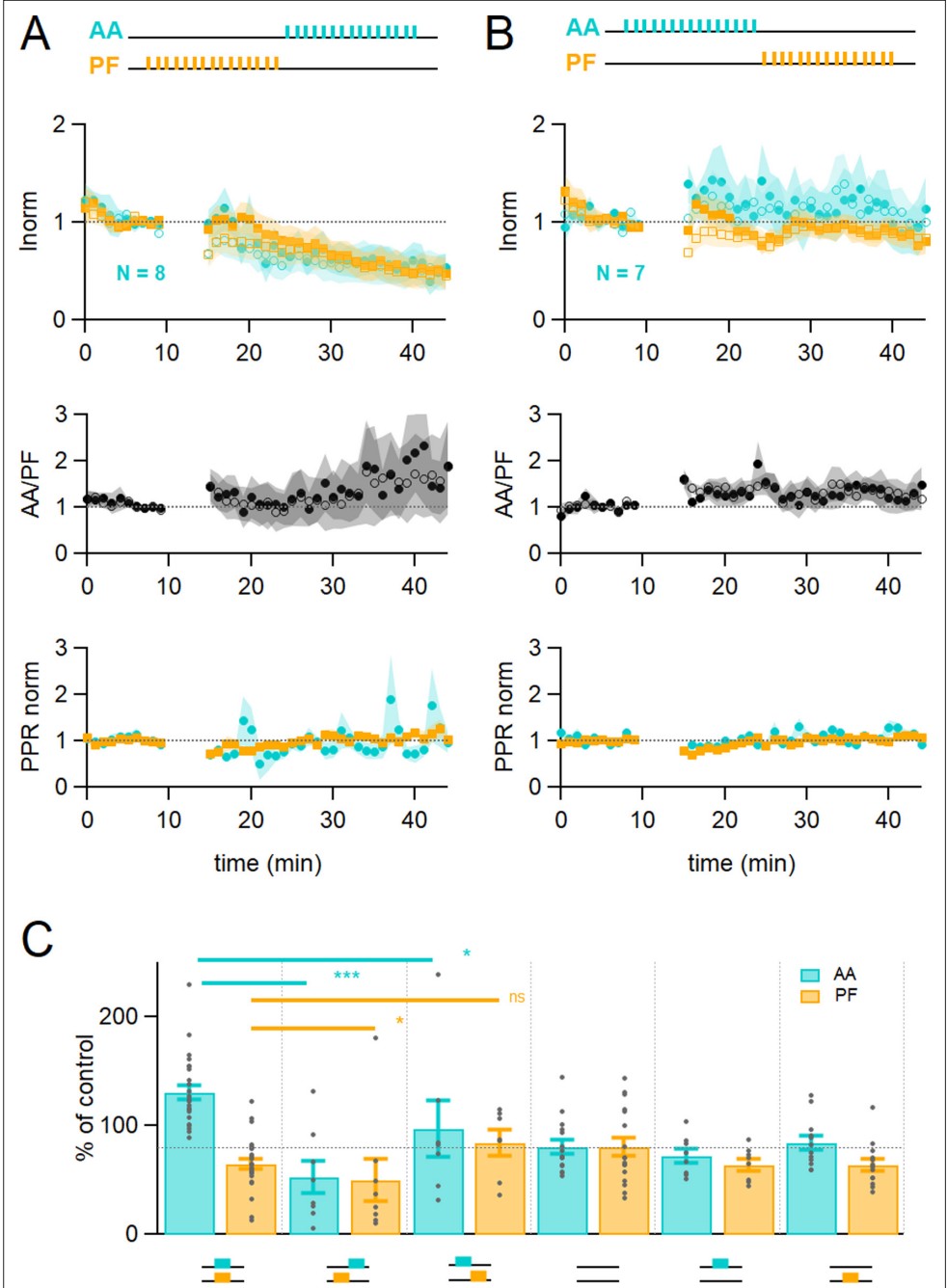

**Figure 2.** Time dependence of plasticity. (**A**) On average, when stimulating ascending axons (AAs) 150 ms after parallel fibres (PFs), the AA- and PF-excitatory postsynaptic currents (EPSCs) decreased by a similar extent (AA was 53 ± 15 %, PF 50 ± 20% of control, n=8, not significantly different from each other p=0.84). These changes were significantly different from experiments with synchronous stimulation (p=8e-5 for AA and p=0.022 for PF). The relative normalised amplitude of the AA pathway increased but not significantly (AA/PF was 224 ± 75% of control, n=7, p=0.2). Colour coding for AA and PF input pathways. Closed symbols for the first and open symbols for the second EPSCs. (**B**) When stimulating AAs 150 ms before PFs, the AA-EPSC facilitated, but declined back to baseline (AA was 97 ± 26% of control, n=7, significantly smaller than synchronous stimulation p=0.011), and the PF-EPSC was maintained close to baseline (PF 84 ± 12% of control, n=7, not significantly larger than synchronous stimulation, p=0.062). The relative normalised amplitude of the AA pathway and the PPR were not affected significantly (AA/PF was 126 ± 31% of control, p=0.35, and $PPR_{AA}$ = 114.4 ± 6.4%, $PPR_{PF}$ = 107.1 ± 4.6%, n=7). (**C**) Average amplitude at 25–30 min as a percentage of baseline for various timing of stimulation, control No Stim, AA only, and PF only stimulation. Individual data points are overlaid for each type of experiment. The AA-EPSC was

*Figure 2 continued on next page*

*Figure 2 continued*

131 ± 7% of baseline (n=24) 25–30 min after induction when AA and PF stimulation was synchronous. It was 53 ± 15% (n=8) when AA stimulation was delayed by 150 ms, and 97 ± 26% (n=7) when PF stimulation was delayed by 150 ms. The PF-EPSC was 65 ± 5% of baseline (n=25) when stimulation was synchronous, 50 ± 20% (n=8) when AA stimulation was delayed by 150 ms, and 84 ± 12% (n=7) when PF stimulation was delayed by 150 ms. The horizontal dotted line indicates EPSC amplitude at the end of the No Stim experiments. (***p<0.001, *p<0.05, ns: not significant). Values given are mean ± SEM. SEM is represented by shading. Wilcoxon Mann Whitney tests.

These experiments show that the relative timing of the AA and PF input stimulation determines the extent of plasticity of both GC synapses, sometimes in opposite directions. The AA-EPSC was either facilitated or depressed significantly when stimulation was synchronous or delayed with respect to PF-EPSC during the induction protocol.

## Plasticity induction mechanism

PF-LTD and LTP have been linked to mGluR1 (*Daniel et al., 1992*; *Konnerth et al., 1992*; *Hémart et al., 1995*) and/or NMDAR activation (*Casado et al., 2002*; *Bidoret et al., 2009*; *Bouvier et al., 2016*; *Kono et al., 2019*), and KOs of either receptor show defects in cerebellar learning (*Aiba et al., 1994*; *Nakao et al., 2019*; *Kono et al., 2019*; *Schonewille et al., 2021*). mGluR1s are highly expressed by PCs and located postsynaptically at GC to PC synapses. mGluR1s activation requires PF train stimulation, and it has not been observed when stimulating sparse GC synapses (*Marcaggi and Attwell, 2005*). NMDARs on the other hand are expressed by molecular layer interneurons (*Glitsch and Marty, 1999*; *Duguid and Smart, 2004*) and GCs (*Bidoret et al., 2009*; *Bidoret et al., 2015*), where they are located on dendrites and presynaptic varicosities. NMDARs are not present postsynaptically at granule cell to Purkinje cell synapses (*Llano et al., 1991*; *Piochon et al., 2007*). We investigated the requirement for mGluR1 and NMDAR activation in the induction of associative plasticity of the AA and PF pathways.

In *Figure 3A and B*, mGluR1s were blocked using the competitive mGluR1 antagonist CPCCOEt (50 µM). *Figure 3A* shows a sample experiment in which both the EPSCs and the delayed IPSCs decreased following induction. In these conditions, on average the AA-EPSC increased transiently following induction, but there was a long-term decrease within 25–30 min (AA was 72 ± 13% of baseline, statistically smaller than control synchronous stimulation, p=0.0004, n=7 and n=24), and the PF-EPSC was also decreased (PF was 54 ± 8%, n=7, not significantly smaller than control synchronous stimulation, p=0.12, n=7). The transient decrease in PPR was present also in these conditions and for both inputs, indicating no involvement of the mGluR1 in this process.

*Figure 3C* shows the effect of inhibiting NMDARs with the NMDAR competitive antagonist APV (50 µM) on associative AA and PF plasticity induction. With NMDARs blocked, AA-LTP observed after synchronous stimulation was completely suppressed. The AA- and PF-EPSCs were on average 50 ± 8% and 40 ± 9% of baseline 25–30 min after induction (n=6, significantly smaller than control p=2e-6, and n=7, p=0.027, respectively). When compared to the PF pathway, depression was not significantly different at AA synapses and there was no relative amplitude change (AA/PF=109 ± 22%, n=5, p=0.34) at 30 min. APV did not affect the transient depression of the PPRs. These data show that NMDAR activation is required for the associative AA-LTP described here.

These results show the concomitant involvement of both mGluR1s and NMDARs in the induction of plasticity at GC synapses.

## Role of inhibition

PCs receive GABAergic inhibition from molecular layer interneurons directly recruited by GCs. GABA$_A$Rs are also present on GCs, including presynaptically on the GC axon (*Stell et al., 2007*). These presynaptic GABA$_A$Rs are activated by the synaptic release of GABA and affect the axonal Cl$^-$ concentration and synaptic release (*Stell et al., 2007*; *Astorga et al., 2015*; *Berglund et al., 2016*). Several recent studies have shown that the plasticity of PF-EPSCs is affected by GABAergic inhibition. *Binda et al., 2016* showed that PF-LTP in mice relies on GABA$_A$R activation to hyperpolarize PC dendrites and allow the recruitment of T-type Ca$^{2+}$ channels. *Rowan et al., 2018* also showed that the recruitment of molecular layer interneuron-mediated inhibition can modify CF-mediated Ca$^{2+}$ signals and LTD induction.

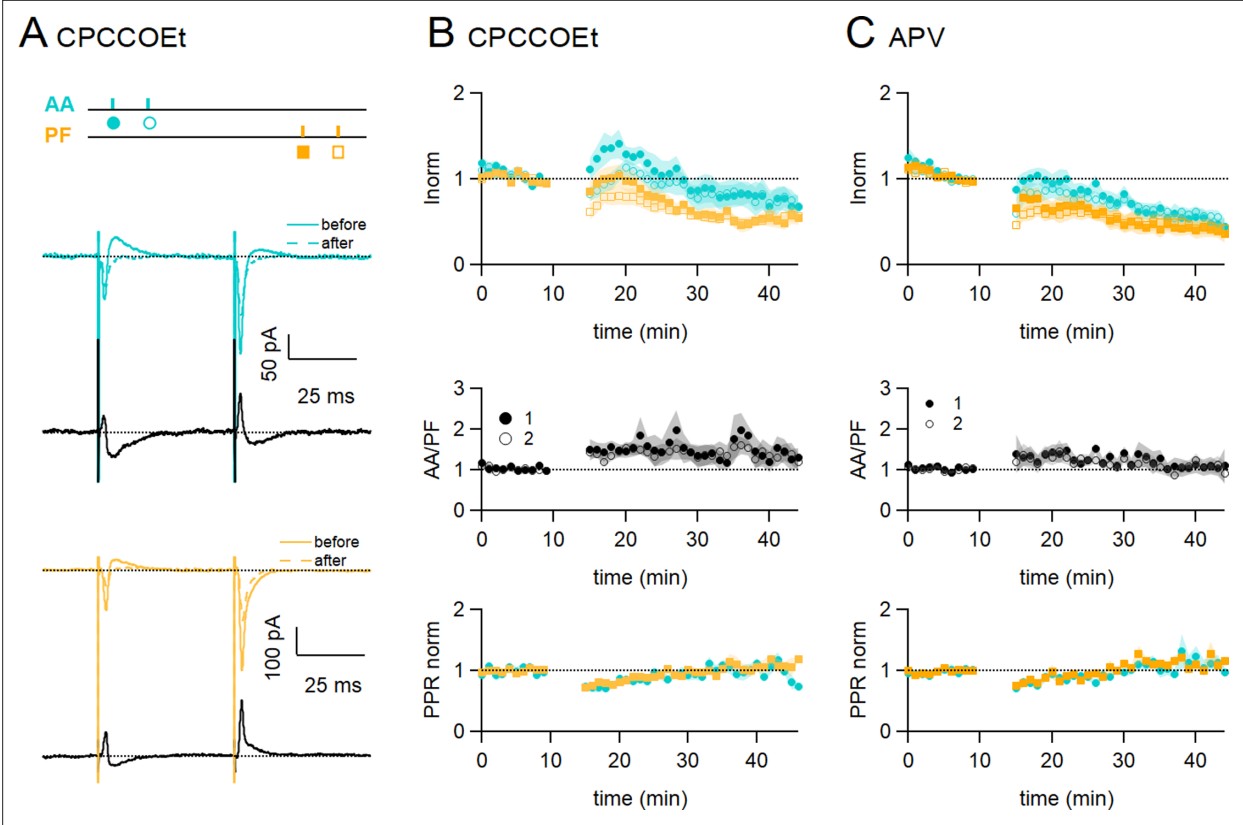

**Figure 3.** Role of NMDARs and mGluR1s. (**A**) Bath application of CPCCOEt (50 µM), a selective blocker of mGluR1Rs, strongly inhibits long-term potentiation (LTP) of ascending axon (AA)-excitatory postsynaptic currents (EPSCs). Sample recordings from one experiment. Traces are the average of the AA (blue) and parallel fibre (PF) (orange) synaptic responses, before (5–10 min, continuous line) and after induction (25–30 min, dotted line). Subtraction traces (25–30 min - 5–10 min) in black. A decrease in the AA-EPSC and PF-EPSC is observed. The test stimulation protocol is depicted with colours and symbols as used in the plots in the following sections for the first (closed) and the second EPSCs (open). (**B**, **Top**) Average time course of the normalised AA- and PF-EPSCs (n=8). mGluR1 receptor block impairs AA-LTP. AA was 72 ± 13% of baseline (statistically smaller than control synchronous stimulation p=0.0004, n=7 and n=24, and the PF-EPSC wass 54 ± 8%, n=7, not significantly smaller than control synchronous stimulation, p=0.12, n=7) (**Middle**) a small sustained increase in the ratio of normalised amplitudes (AA/PF) is observed. (**Bottom**) Plot of the normalised PPR of both inputs. (**C**, **Top**) Average time course of the normalised AA- and PF-EPSCs (colours and symbols as in A) in the presence of APV (50 µM) (n=7). Both AA and PF pathways are depressed, showing that NMDAR activation is necessary for AA-LTP induction. The AA- and PF-EPSCs were on average 50 ± 8% and 40 ± 9% of baseline 25–30 min after induction (n=6, significantly smaller than control p=2e-6, and n=7, p=0.027, respectively)(**Middle**) The ratio of normalised amplitudes (AA/PF) is slightly increased, reflecting a slower depression of the AA inputs. (**Bottom**) Plot of the normalised PPR of both inputs. Values given are mean ± SEM. SEM is represented by shading. Wilcoxon Mann Whitney tests.

The role of GABA$_A$Rs-mediated inhibition was tested during plasticity induction. To this end, a series of experiments was performed with the GABA$_A$R antagonist SR95531 (3 µM) in the bath throughout the recording, and the AA and PF pathways were stimulated simultaneously (*Figure 4*). The sample experiment in *Figure 4A* shows that the AA-EPSC was slightly increased, while the PF-EPSC was decreased. On average with GABA$_A$Rs blocked, the AA-EPSC was initially facilitated after induction (170 ± 30%, n=9), but that potentiation decreased with time and it was 117 ± 24% of baseline after 25–30 min. The outcome of induction on the AA pathway was highly variable, and in 2 out of 9 cells, the AA-EPSC was strongly depressed at 25–30 min. On average, AA-LTP was smaller than in control experiments, but given the variability observed, it was not significantly different from control experiments (p=0.1). The decrease of the PF-EPSC at 25–30 min (56 ± 10%, n=9) was not significantly smaller than control values (p=0.19). The relative plasticity of AA and PF inputs was stable following induction (AA/PF=330 ± 130% at 25–30 min after induction n=9). The PPR of both pathways was transiently decreased. GABA$_A$R activation, therefore, seems to be required for efficient induction and maintenance of this type of associative plasticity.

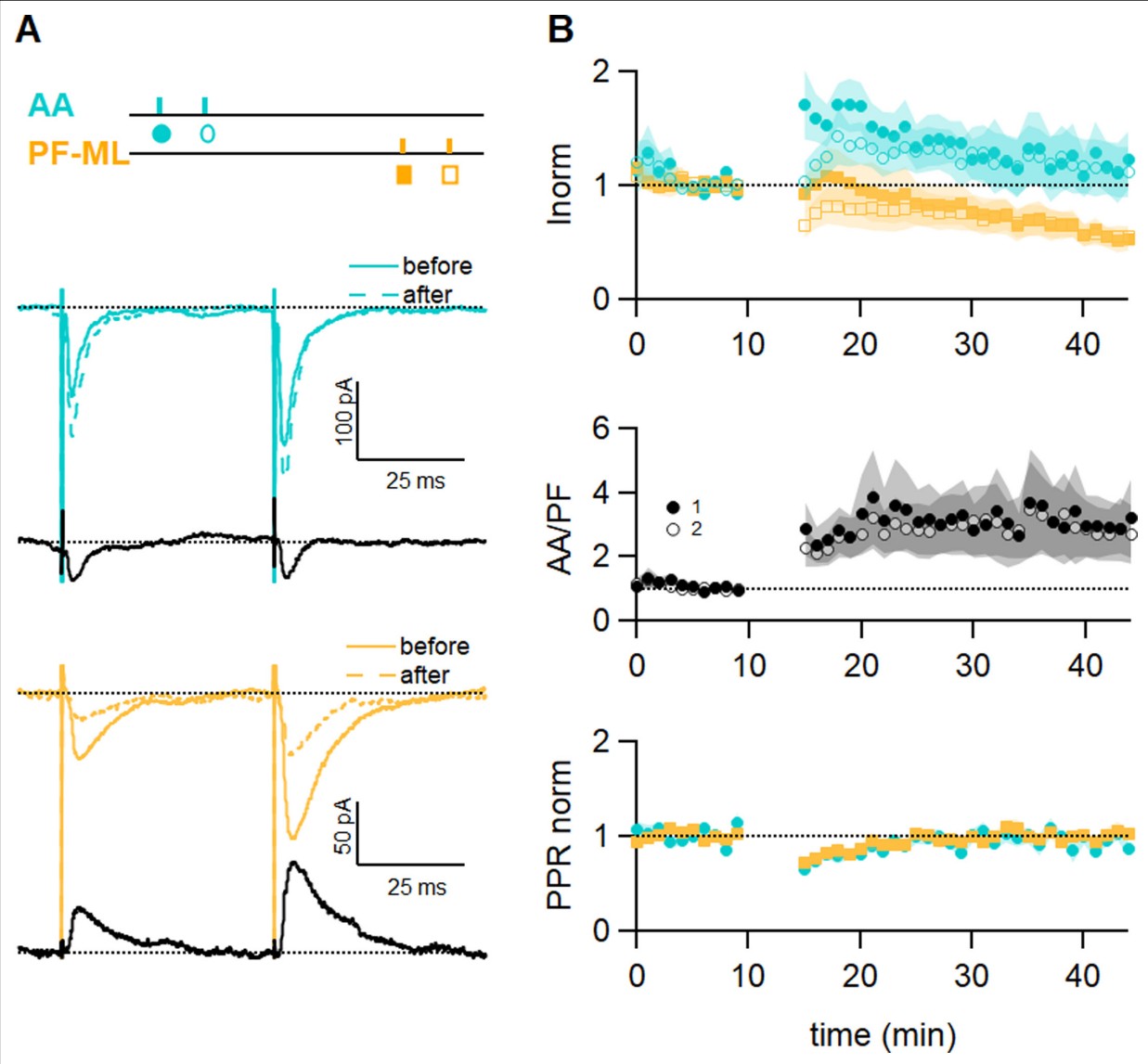

**Figure 4.** GABA$_A$ receptor block affects long-term plastic changes. (**A**) Sample recordings from an experiment in the presence of the GABA$_A$R antagonist SR (3 μM). Paired excitatory postsynaptic currents (EPSCs) evoked before and after induction, together with subtraction traces (black), are shown for the ascending axon (AA) (blue) and parallel fibre (PF) (orange) pathways. The AA-EPSC is increased after induction and the PF-EPSC is strongly decreased in this experiment. The test stimulation protocol is depicted with colours and symbols as used in the plots in the following section for the first (closed) and the second EPSCs (open). (**B**) Average time course of the normalised AA- and PF-EPSCs (n=9). The big errors in AA peak amplitude observed after induction are due to the large variability of the outcome of the protocol on this pathway (AA 117 ± 24%; PF 56 ± 10%, n=9; not significantly different from control experiments p=0.1 and not significantly smaller than control values p = 0.19 respectively, Wilcoxon Mann Whitney tests). The ratio of the normalised AA and PF amplitude shows the same variability, while the normalised PPR displays the same relative error and time course as control experiments. Values given are mean ± SEM. SEM is represented by shading.

## Role of synaptic inputs distribution

We have shown that paired stimulation of AA and PF triggers AA-LTP. AAs can make several synapses per axon, and these are sparsely distributed on the vertical axis of the PC dendritic tree. On the other hand, because PFs are stimulated in the molecular layer, they are stimulated as a dense beam of fibres and, therefore, a dense patch of synapses. We asked whether the induction of AA-LTP is due to the sparse distribution of AA synapses rather than the specific properties of AA inputs.

To test for the role of the sparse distribution of the inputs, a series of experiments was performed substituting local AA stimulation with GC layer stimulation at a distance of 100–180 μm from the recorded PC. In this configuration, GC layer stimulation recruits PF synapses sparsely distributed

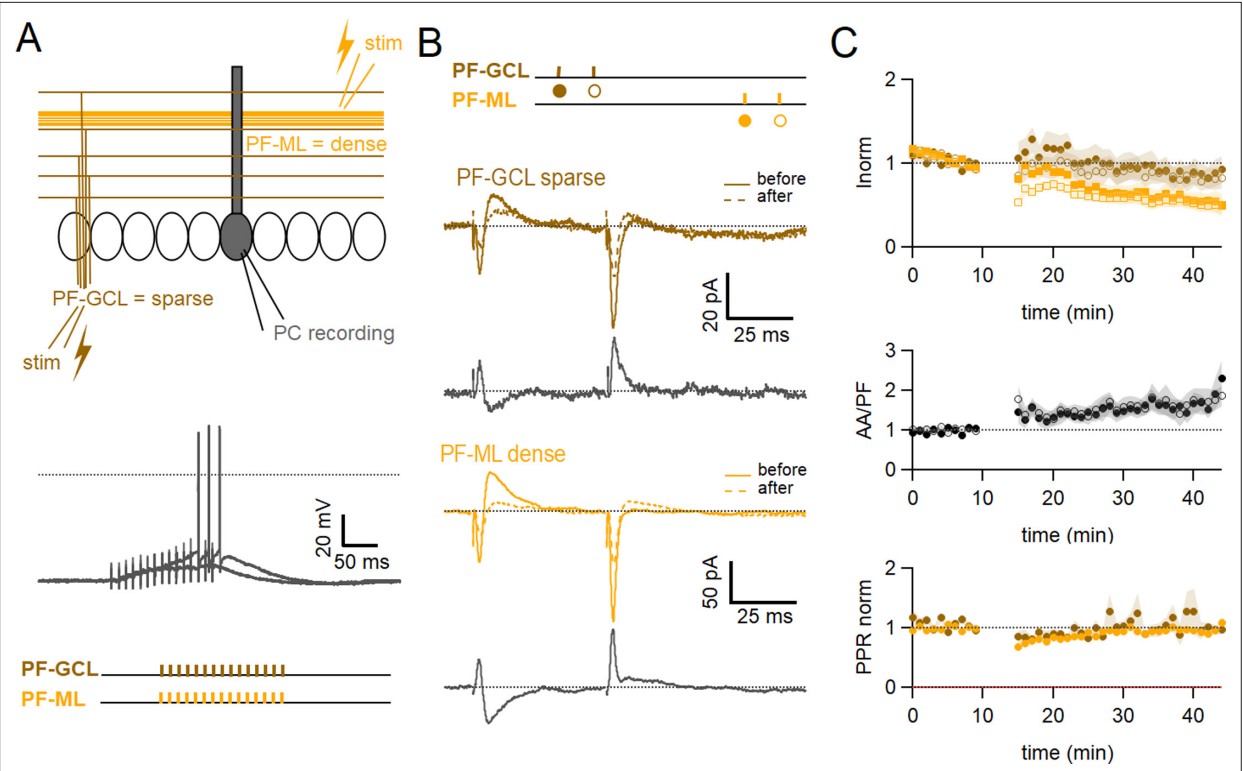

**Figure 5.** Role of the sparse input distribution. (**A**) To test the role of the sparse input distribution, a stimulation electrode was positioned in the granule cell (GC) layer, stimulating only parallel fibre (PF) synapses sparsely distributed on the PC dendrites. The induction protocol was applied to the sparse PF pathway (PF-sparse), together with the dense PF pathway stimulated in the molecular layer (PF-dense). (**B**) Example experiment. Traces show the PF-sparse and -dense excitatory postsynaptic currents (EPSCs) before and after induction, and subtractions (grey) to highlight changes. The test stimulation protocol is depicted with colours and symbols as used in the plots in the following section for the first (closed) and the second EPSCs (open). 25–30 min after induction, both the sparse and dense EPSCs are reduced. (**Bottom left**) IClamp responses to the first two trains of stimulation. (**C**) Average time course of the normalised EPSC amplitudes (n=10). Both sparse and dense inputs are reduced on average to 88%±13% (significantly smaller than control AA, p=0.0024, n=10 PF-sparse and n=24 AA control) and 54 ± 10% of baseline (significantly smaller than control PF, p=0.017, n=10 PF-dense and n=25 PF control) after 25–30 min, respectively. With time, the dense PF input was more strongly depressed, and the ratio of normalised amplitudes (PF-sparse/PF-dense) increased to 182%±24% of control. Values given are mean ± SEM. SEM is represented by shading. Wilcoxon Mann Whitney tests.

on the PC dendrites and cannot involve AA synapses. The induction protocol was then applied to the sparse PF pathway (PF-sparse), together with the dense pathway stimulated in the molecular layer (PF-dense) (see *Figure 5A*). *Figure 5B* shows an example of an experiment. Traces show the PF-sparse and -dense EPSCs before and after the induction protocol and subtractions (grey) to highlight changes. 25–30 min after induction, both the sparse and dense EPSCs are reduced. *Figure 5C* shows the average time course of 10 experiments. Both the sparse and dense inputs are reduced on average to 88 ± 13% of baseline (significantly smaller than control AA, p=0.0024, n=10 PF-sparse and n=24 AA control) and 54 ± 10% of baseline (significantly smaller than control PF, p=0.017, n=10 PF-dense and n=25 PF control) after 25–30 min, respectively. When AA stimulation was replaced by sparse PF stimulation, the amplitude of the PF-sparse input was not increased as seen for AA inputs, but depressed. There was initially no change in the ratio of normalised amplitudes of the two pathways, although with time, the dense PF input stimulated in the molecular layer was more strongly depressed than the sparse input, and the relative plasticity of PF-sparse increased (PF-sparse/PF-dense=182 ± 24%).

These experiments suggest that associative LTP of the AA input is not due to the sparse distribution of stimulated AA synapses on PC dendrites, but to the stimulation of a population of synapses prone to a specific form of plasticity.

## Discussion

We show for the first time that stimulating AA and PF inputs simultaneously triggers LTP of the stimulated AA inputs specifically. AA-LTP is associative as it is not observed with stimulation of either pathway alone and it is timing-dependent. AA-LTP is weakly dependent on GABAergic inhibition, but most interestingly also, it depends on the activation of both mGluR1s and NMDARs. It is linked to the identity of AA synapses rather than their sparse distribution on PC dendrites, as it is not observed for sparse PF inputs. Below we discuss these findings and potential consequences for cerebellar physiology.

### Identification of AA synapses

We have shown that the difference in plasticity observed between the AA and PF inputs is not linked to the distribution of synapses on the dendritic tree but to the different properties of the synapses. We have argued earlier that our approach for stimulating fibres recruits mainly PF synapses, for molecular layer stimulation, and mainly AA synapses, for local GC layer stimulation. However, we have no morphological identification to verify that all stimulated synapses are strictly AA or PF synapses. Strictly speaking, GC layer stimulation close to the PC soma will recruit synapses from local GCs, formed by the initial portion of the axons, including some formed by the PF close to the bifurcation point, whereas molecular layer stimulation will recruit mostly synapses from distal GCs and formed by the distal portion of the axons. The observed associative plasticity could thus alternatively be due to a gradient of properties of synapses from proximal to distal sites along the GCs axon. If there is a gradient however, it is limited to short distances compared to the total fibre length, as this effect was not observed when stimulating PFs in the GC layer 100–180 μm from the recorded cell (4 to 5% of the total fibre length, *Figure 5*). If the plasticity described here applies to a gradient along the GC axon rather than to AAs and PFs, this would not impact the mechanisms identified nor the consequences for cerebellar physiology.

### AA plasticity

We have observed AA-LTP with synchronous stimulation and AA-LTD when AAs were stimulated 150 ms after PFs (*Figure 2*). To our knowledge, only Sims and Hartell (2005, 2006) examined plasticity at AA synapses and reported the absence of both LTP and LTD associated with CF conjunctive stimulation.

For both AA and PF pathways, the PPR transiently decreased following the induction protocol. Changes of the PPR have been linked to a change in the presynaptic release probability (*Zucker and Regehr, 2002*). This suggests a transient presynaptic effect following induction, which likely explains the progressive changes in AA and PF responses. Presynaptic GABA$_B$ and CB1 receptors have been shown to modulate the presynaptic release probability at granule cell to Purkinje cell synapses, but they decrease release and increase the PPR (Dittman and Regehr, 1997; *Safo and Regehr, 2005*). Also, endocannabinoid release is triggered by mGluR1 activation, but the mGluR1 block did not affect the transient decrease of the PPR. Therefore, activation of these receptors cannot account for the transient presynaptic effect we observed. However, there was no significant long-term change in the PPR, suggesting that AA-LTP is due to postsynaptic changes.

Activation of both mGluR1s and NMDARs was required for AA-LTP induction (see *Figure 3*). This is interesting in view of the past involvement of these receptors, mostly independently, in postsynaptic forms of cerebellar PF LTD and LTP, both linked to PF stimulation (*Daniel et al., 1992*; *Konnerth et al., 1992*; *Hémart et al., 1995*; *Lev-Ram et al., 1997*; *Casado et al., 2002*; *Safo and Regehr, 2005*; *Bidoret et al., 2009*; *Bouvier et al., 2016*). The difference with previous reports lies with the simultaneous stimulation of AA with a PF beam. The conjunctive requirement for NMDAR and mGluR1 activation could be explained based on the dependence of plasticity on the concentrations of both NO and postsynaptic calcium (*Coesmans et al., 2004*; *Safo and Regehr, 2005*; *Bouvier et al., 2016*; *Piochon et al., 2016*). NO production has been linked to the activation of NMDARs in granule cell presynaptic boutons (*Casado et al., 2002*; *Bidoret et al., 2009*; *Bidoret et al., 2015*; *Bouvier et al., 2016*), or occasionally in molecular layer interneurons (*Kono et al., 2019*). NO is a diffusible messenger and it activates the Guanylate Cyclase in the postsynaptic Purkinje cell. On the other hand, mGluR1s in Purkinje cells are strongly linked to Ca$^{2+}$ signalling. They activate several downstream pathways, including IP$_3$-mediated Ca$^{2+}$ release from stores (*Finch and Augustine, 1998*; *Takechi et al., 1998*), a slow EPSC mediating calcium entry (*Canepari et al., 2004*; *Canepari and Ogden, 2006*) and regulate

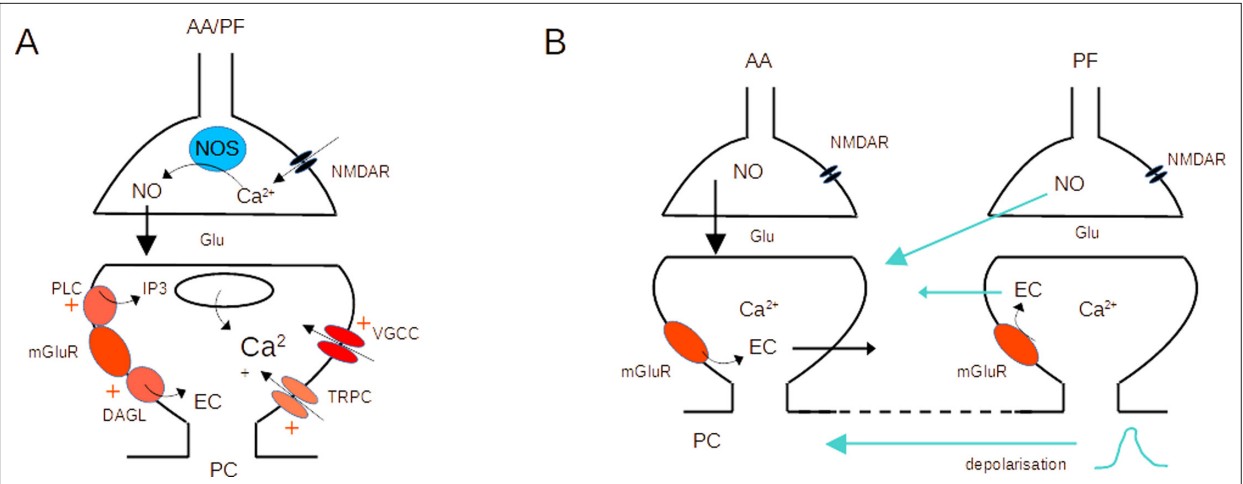

**Figure 6.** Molecular pathways and potential crosstalk mechanisms involved in AA-long-term potentiation (LTP). (**A**) Activation of presynaptic NMDARs triggers Ca²⁺ entry, activation of NOS, and production of NO. NO diffuses to the postsynaptic compartment where it triggers Guanylate Cyclase activation and regulation of AMPAR number. Activation of postsynaptic mGluR1s in the PC activates PLC, DAGL, TRPC, and a modulation of VGCC, triggering an increase in the Ca²⁺ concentration and EC production. NO and Ca²⁺ concentration control plasticity (**Bouvier et al., 2016**). (**B**) The dendritic depolarisation triggered by activation of the dense patch of PF synapses might prime Ca²⁺ release from stores linked to mGluR1 activation. Alternatively, NO or EC produced at PF synapses might diffuse to AA synapses. Potential sources of crosstalk are highlighted in blue. AA ascending axon, DAGL Diacylglycerol lipase, EC Endocannabinoids, Glu glutamate, NO Nitric Oxide, NOS Nitric Oxide Synthase, PC Purkinje cell, PF parallel fibre, PLC phospholipase C, TRPC Transient receptor potential canonical channel, VGCC voltage-gated calcium channels.

voltage-activated Ca²⁺ channels (**Hildebrand et al., 2009**; **Otsu et al., 2014**; **Khawaled et al., 1999**; **Ait Ouares and Canepari, 2020**), all leading to an increase of the postsynaptic calcium concentration. NMDARs and mGluR1s could cooperate to control postsynaptic plasticity (see **Figure 6A**).

Presynaptic NMDAR and postsynaptic mGluR1 activation could take place at AA and PF synapses independently. However, the need for coincident activation of the two synaptic inputs suggests either spatial overlap or a requirement for crosstalk between the two synaptic territories. Our experiments used horizontal slices, where PC dendrites dip into the slice, so we have no indication of whether the two synaptic territories overlap. Given the size of the synaptic currents (**Figure 1—figure supplement 2**) however, they represent a small number of synapses compared to the total number formed by granule cells on a PC. The median PF-EPSC (288 pA, n=25) corresponds to approximately 34 active synapses (unitary amplitude for distal GC to PC connection is 8.4 pA **Isope and Barbour, 2002**). Taking into account that 85% of the synapses formed by PFs onto GCs are silent, our stimulation likely recruits about 230 synapses, active and silent, put together. For a similar unitary synaptic size, the median AA-EPSC (90 pA) would correspond to 11 active synapses and a total of 22 synapses (50% of local GC to PC connections are silent **Isope and Barbour, 2002**). These numbers represent less than 0.2% of the 175,000 excitatory synapses present on the dendritic tree of a PC (**Napper and Harvey, 1988**). Given that we placed the PF stimulating electrode at variable depths and distances from the recorded Purkinje cell, there is statistically little chance of reproducible overlap between the territories of PF and AA recruited. Moreover, AA synapses will be spread in the vertical axis of the dendrites whereas PF synapses will form a dense patch on the PC dendrites. Spatial overlap is, therefore, unlikely to happen in a significant proportion of experiments, suggesting that for interaction to occur a signal needs to propagate from PF to AA synapses to explain associative AA-LTP.

Crosstalk might be required because a receptor is absent from AA synapses or a pathway fails to activate. The specific localisation of receptors at AA vs PF synapses is unknown. If mGluR1 is present at AA synapses, stimulation could activate the mGluR1 slow EPSC or mGluR1 mediated presynaptic endocannabinoid inhibition. However, this is not observed at sparse PF inputs unless glutamate transporters are blocked (**Marcaggi and Attwell, 2005**), which is thought to reflect insufficient glutamate buildup, and makes it difficult to localise the receptors. mGluR1 mediated Ca²⁺ release from stores shows a higher sensitivity to glutamate (**Canepari and Ogden, 2006**) and might take place with sparse inputs, but Ca²⁺ signals have not been investigated in this configuration, and we do not know

if mGluR1s are present at AA synapses. Similarly, we do not know whether presynaptic NMDARs are present.

A crosstalk mechanism would require the propagation of a signal between synaptic territories. mGluR1-mediated $Ca^{2+}$ signals in Purkinje cells are not expected to propagate across the PC dendritic tree, because both $Ca^{2+}$ and $IP_3$ are quickly buffered. On the other hand, mGluR1-mediated $Ca^{2+}$ release from stores is known to require priming by depolarisation (*Canepari and Ogden, 2006*). It is possible that mGluR1s are present at AA synapses, but $Ca^{2+}$ release fails to activate because the local depolarisation is insufficient. The PF beam stimulated forms a dense patch of synapses which will be very efficient at depolarising the membrane potential locally, and this depolarisation might spread to AA synapses if it reaches the threshold for active conductances. Fluctuations of the dendritic membrane potential triggered by the PF-EPSC might enable mGluR1-mediated $Ca^{2+}$ signals at AA synapses. The isolation of AA synapses in fine dendritic branches (*Lu et al., 2009*) might make them more prone to depolarisation. mGluR1 activation also triggers the production of endocannabinoids (EC; *Maejima et al., 2001*; *Brown et al., 2003*), and presynaptic CB1Rs have been involved in PF-LTD (*Safo and Regehr, 2005*; *Carey et al., 2011*). Similarly, NO is released by the activation of presynaptic NMDARs. EC and NO are both diffusible messengers and potential substrates for crosstalk between AA and PF synapses. Activation of PF synapses might be required for NO or EC production and diffusion to AA synapses (See *Figure 6B* for potential crosstalk mechanisms). Further experiments will be required to test whether EC or NO are sufficient to induce AA-LTP when stimulating AA inputs on their own.

## PF synapses

In the experimental conditions used here, the induction protocol had no significant effect on the amplitude of the PF-EPSC although we observed a tendency to decrease. This was unexpected because the induction protocol we used, when applied to PF only and with inhibition preserved, was previously shown to induce PF-LTP (*Binda et al., 2016*; *Jörntell and Ekerot, 2002*), although not always (*González-Calvo et al., 2021*). But *Binda et al., 2016* also showed that PF-LTP was not sustained when inhibition was blocked during induction, contradicting other studies performed with inhibition blocked (*Bouvier et al., 2016*; *Piochon et al., 2016*; *Schonewille et al., 2021*) with Picrotoxin or Bicuculline (a blocker of SK channels, *Khawaled et al., 1999*). Moreover, PF-LTP either relied on mGluR1 activation (*Binda et al., 2016*), or NMDAR activation (*Bidoret et al., 2009*; *Bouvier et al., 2016*; *Schonewille et al., 2021*). Therefore, we do not have an explanation for the lack of PF-LTP, but the literature shows multiple contradictions, likely indicating that the processes involved are sensitive to the experimental conditions. Noticeable experimental differences with the study of *Binda et al., 2016* are the use of mice rather than rats, differences in the orientation of cut of the slices, which might have resulted in the study of different cerebellar lobules, but importantly also, we used a low internal $Cl^-$ concentration, reproducing the concentration estimated in PCs (*Chavas and Marty, 2003*), and the recording solution was supplemented with 10 µM of the $Ca^{2+}$ buffer EGTA.

## Role of inhibition

Experiments were performed with inhibition preserved and $GABA_A$Rs activation was shown to secure efficient induction of AA-LTP (see *Figure 4*). Block of $GABA_A$Rs did not suppress the initial increase in amplitude, but under these conditions, plasticity was not sustained, highly variable, and not significant when compared to control experiments. Inhibition indeed seems to interfere with plasticity induction (*Binda et al., 2016*; *Rowan et al., 2018*). At this stage, we have no detail on the mechanism of influence of GABAergic inhibition in AA-LTP induction. *Binda et al., 2016* showed that hyperpolarisation by IPSCs relieves inactivation of T-type $Ca^{2+}$ channels, themselves regulated by mGluR1, boosting $Ca^{2+}$ signals and triggering PF-LTP. That mechanism would seem compatible with our results at AA synapses although we do not know whether it requires NMDARs activation. On the other hand, the work by *Rowan et al., 2018* indicated that inhibition decreased $Ca^{2+}$ entry, and as a result converted LTD into LTP. In this study, inhibition was recruited using optogenetics rather than synaptic excitation, and it is possible that the pattern of inhibition generated was less physiological, both with respect to the number of molecular layer interneurons stimulated and timing. This might explain the different results.

## Cerebellar physiology

We argued in the introduction that, as a consequence of morphology, AAs can only form synapses with a few PCs, while PFs course through and synapse with the dendrites of hundreds of PCs. AA synapses are suited to encode precise and selective information from the receptive field, while PF synapses could efficiently represent context. This work is consistent with the idea that they are functionally different and might fulfil distinct roles with respect to the computation performed by the cerebellar cortex. The associative form of plasticity we describe, together with the time dependence, is expected to reinforce AA signals presented conjointly with PF, or within a pertinent time window, while attenuating signals presented in isolation. This would lead to the strengthening of the inputs linked to activity in other receptive fields.

This work shows that associative plasticity in PCs can be triggered by granule cell inputs independently of the climbing fibre, requiring only moderate inputs, in contrast with the powerful CF, which triggers widespread depolarisation and $Ca^{2+}$ increase. The climbing fibre is classically associated with error signals and learning (*Aiba et al., 1994*; *Ito and Kano, 1982*; *Marr, 1969*; *Medina and Lisberger, 2008*; *Yang and Lisberger, 2014*). However, both climbing fibre (*Kostadinov et al., 2019*) and granule cells *Wagner et al., 2017* have been shown to encode reward signals, which might be important in driving plasticity.

# Methods

## Ethical approval

Sprague Dawley rats were provided by Janvier (St Berthevin, France) or bred and subsequently housed at the Animal Housing and Breeding facility of BioMedTech facilities (INSERM US36, CNRS UAR2009, Université Paris Cité) in agreement with the European Directive 2010/63/UE regarding the protection of animals used for experimental and other scientific purposes. Experimental procedures were approved by the French Ministry of Research and the ethical committee for animal experimentation at Paris Cité.

## Slice preparation

Experiments were performed on horizontal slices 300 µm thick cut from the cerebellum of 19–25- day-old Sprague-Dawley male or female rats. Briefly, rats were killed by decapitation under general anaesthesia following inhalation of the volatile anaesthetic isoflurane at a concentration of 3–4% in accordance with the Directive 2010/63/UE, and the cerebellum was quickly removed. After the removal of the brainstem, the tissue was glued to the stage of a vibratome (Leica VT1200S, Germany). Slices were cut at a temperature of 34 °C and subsequently kept in a vessel bubbled with 95% $O_2$/5% $CO_2$ at this temperature. Slice preparation and recordings were made in a bicarbonate buffered solution containing in mM: 115 NaCl, 2.5 KCl, 1.3 $NaH_2PO_4$, 26 $NaHCO_3$, 2 mM $CaCl_2$, 1 mM $MgSO_4$, 0.1 mM Ascorbic Acid, and 25 glucose.

## Patch-clamp recording of synaptic currents

Whole-cell patch-clamp recordings were made from PCs, identified by their size and location at the edge of the molecular and GC layers, with an EPC10 amplifier (HEKA, Germany) and PatchMaster acquisition software. Bath temperature was kept at 30–32°C The internal solution contained in mM: 135 KGluconate, 10 $K_2$ Creatine Phosphate, 10 HEPES, 0.01 EGTA, 2.5 $MgCl_2$, 2 $ATPNa_2$, and 0.4 GTPNa, pH adjusted to 7.3 with KOH and osmolarity to 295 mOsm/kg. When filled with the internal solution, recording pipettes had a resistance between 3 and 4 MΩ. Membrane currents were recorded at a pipette potential of –60 mV (not corrected for junction potential of approximately –12 mV pipette-bath). Series resistance was 80–90% compensated. During experiments, the preparation was visualised on an upright microscope (Olympus BFX51; 60 × 0.9 NA water dipping objective) and the bath was continuously perfused at a rate of 5 ml/min (5 bath volumes per min) with solution equilibrated with 95% $O_2$/5% $CO_2$ to maintain pH and solution recirculated.

Plasticity of GC to PC synapses was studied in horizontal cerebellar slices as these slices better preserve PFs running in the plane of cut. At this age, GC synapses are well-established (*Ichikawa et al., 2016*). AAs and PFs were stimulated with patch pipettes slightly larger than those used for recordings filled with a Hepes-buffered external solution and positioned either in the molecular (lower

two-thirds) or the GC layer, as discussed in the result section. The baseline amplitude of both AA and PF pathways were sampled with a pair of suprathreshold pulses at 50 ms intervals, delivered every 10 s (*Figure 1A* top panel, stimulation was biphasic, 100–180 µs duration, 5–15 V). The AA and PF test stimuli were separated by 0.5 s. Stimulation strength was adjusted between 5 and 15 V to stimulate AA- and PF-EPSCs reliably. PF stimulation was usually very efficient as fibres are densely packed in the molecular layer and PF-EPSC amplitude typically increased smoothly with stimulation intensity. Although we originally worked with large amplitude PF-EPSCs (up to about 1nA), we aimed at amplitudes of a few hundred pA, closer to the amplitude of stimulated AA-EPSCs, by adjusting stimulation intensity. Granule cell somas and axons are more sparse in the granule cell layer and stimulation intensity was adjusted to activate EPSCs a few hundred pA when possible, but increasing stimulation intensity did not necessarily recruit larger EPSCs. Care was taken to avoid stimulating the local Purkinje cell axon and climbing fibre. Once a stable input was obtained, stimulation intensity was raised by one Volt to increase reliability. In most of the experiments, AAs were tested first to avoid possible interference from mGluR1s activation and the release of endocannabinoid by PFs stimulation. No antagonist of the inhibitory inputs was applied. Evoked responses consisted of excitatory postsynaptic currents (EPSCs) that were often quickly followed by inhibitory postsynaptic currents (IPSCs) (mixed EPSC/IPSC, see Appendix 1). The low $Cl^-$ concentration in the intracellular solution ensured the reversal potential for $Cl^-$ was close to the value determined for PCs (–85 mV, *Chavas and Marty, 2003*), and IPSCs were outward at the recording potential of –60 mV. After recording a baseline period of at least 10 min, we applied a stimulation protocol aimed at inducing plasticity. The recording configuration was switched to Current clamp and the potential was set and maintained near –65 mV. The protocol applied (see *Figure 1A* bottom panel) consisted of the synchronous stimulation of both inputs by a train of 15 pulses at 100 Hz repeated every 3 s a total of 100 times. Following the induction protocol, the recording configuration was returned to Voltage clamp, and the test of alternate AA- and PF-EPSC amplitude resumed.

Data were excluded from analysis if synaptic stimulation became unstable and resulted in loss of the synaptic response or unstable responses. In the result section the number of cells sometimes differ for the two pathways for a given set of experiments, because one of the inputs was sometimes lost during the 30 min following induction. Because the protocol was properly applied, the data for the remaining input were included in the results. Occasionally, stimulation of an input started stimulating the Climbing fibre or the Purkinje cell itself during the course of an experiment. Climbing fibre stimulation precludes measurement of the EPSC and could trigger plasticity. Stimulation intensity of that input was reduced to avoid recruiting the fibre or Purkinje cell, and the input was excluded from the analysis.

## Analysis of evoked EPSCs

For the analysis of synaptic currents, raw current traces were exported to Igor Pro (Wavemetrics), and peak excitatory current amplitudes were measured as the minimum of the synaptic response (mixed EPSC/IPSC) over a time window overlapping the peak and spanning a few sampling points of the average EPSC. Since the plasticity protocol might affect EPSCs and IPSCs differently, and might, therefore, affect our estimate of the peak EPSC and its long-term changes, we conducted a set of experiments to measure the IPSCs and EPSCs separately and confirmed that measuring the minimum of the mixed EPSC/IPSC gives a good estimate of EPSC amplitude and its long term changes (see Appendix 1).

## Chemicals

2-(3-Carboxypropyl)–3-amino-6-(4methoxyphenyl)pyridazinium bromide (SR 95531), D-(-)–2-Amino-5-phosphonopentanoic acid (D-AP5) and 7-(Hydroxyimino)cyclopropa[b]chromen-1a- carboxylate ethyl ester (CPCCOEt) were purchased from Tocris Bioscience or from HelloBio (UK). CPCCOEt was dissolved in DMSO at a concentration of 100 mM. All other stocks were prepared in water. Drugs were diluted in saline just before use. All other chemicals were purchased from Sigma.

## Statistics

Statistical significance was tested with non-parametric methods for most of the data sets. These do not require assumptions about the nature of the distribution of the variables (as parametric

tests do); we used either the Wilcoxon signed rank test (non-parametric, for paired samples) or the Wilcoxon Mann Whitney test (non-parametric, for unpaired samples). The T-test was used for control data only, where N was big enough to show a normal distribution of the variables (see *Figure 1—figure supplement 2*). Tests were conducted using Igor Pro (Wavemetrics). All values given are mean ± SEM.

## Acknowledgements

We would like to dedicate this work to the memory of David Ogden, who was influential in placing the cerebellum at the centre of this work. We thank Brandon Stell and Alain Marty for the discussions on the manuscript. We thank the Animal Housing and Breeding facility and the Prototyping facility of BioMedTech facilities (INSERM US36, CNRS UAR2009, Université Paris Cité) for providing the animals used in this study and assistance with technical developments. The work was funded by ANR grants ANR-19-CE37-011-01 SpinoCereLoco and ANR-18-CE16-0010-01 RewardInhib.

## Additional information

### Funding

| Funder | Grant reference number | Author |
|---|---|---|
| Agence Nationale de la Recherche | ANR-19-CE37-011-01 SpinoCereLoco | Céline Auger |
| Agence Nationale de la Recherche | ANR-18-CE16-0010-01 RewardInhib | Céline Auger |

The funders had no role in study design, data collection and interpretation, or the decision to submit the work for publication.

### Author contributions

Rossella Conti, Conceptualization, Data curation, Formal analysis, Investigation, Methodology, Writing – original draft, Writing – review and editing; Céline Auger, Conceptualization, Data curation, Formal analysis, Funding acquisition, Validation, Investigation, Methodology, Writing – original draft, Project administration, Writing – review and editing

### Author ORCIDs

Rossella Conti ⓘ https://orcid.org/0000-0003-2175-2087
Céline Auger ⓘ https://orcid.org/0009-0003-0203-0297

### Ethics

Sprague Dawley rats were housed at the Animal Housing and Breeding facility of BioMedTech facilities (INSERM US36, CNRS UAR2009, Université Paris Cité) in agreement with the European Directive 2010/63/UE regarding the protection of animals used for experimental and other scientific purposes. Experimental procedures were approved by the French Ministry of Higher Education and Research and the ethical committee for animal experimentation of Paris Descartes (APAFIS#8725-2016100714582450v2).

Reviewer #1 (Public Review): https://doi.org/10.7554/eLife.96140.3.sa1
Reviewer #2 (Public Review): https://doi.org/10.7554/eLife.96140.3.sa2
Reviewer #3 (Public Review): https://doi.org/10.7554/eLife.96140.3.sa3
Author response https://doi.org/10.7554/eLife.96140.3.sa4

## Additional files

### Supplementary files
• MDAR checklist

## Data availability

Raw electrophysiological data for all figures and some of the analysis data are available on Dryad https://doi.org/10.5061/dryad.zs7h44jk0.

The following dataset was generated:

| Author(s) | Year | Dataset title | Dataset URL | Database and Identifier |
|---|---|---|---|---|
| Auger C, Conti R | 2024 | Associative plasticity of granule cell inputs to cerebellar Purkinje cells | https://doi.org/10.5061/dryad.zs7h44jk0 | Dryad Digital Repository, 10.5061/dryad.zs7h44jk0 |

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

## Appendix 1

### Analysis of the excitatory and inhibitory components of the mixed response shows no bias due to overlapping inhibition

Our experiments were performed with inhibition intact and with a Cl$^-$ reversal potential close to the value estimated in PCs (–85 mV, *Chavas and Marty, 2003*). Because GCs also synapse on molecular layer interneurons, most of the time AA and PF stimulation recruits monosynaptic excitation and disynaptic inhibition to PCs, generating a mixed synaptic response with excitatory and inhibitory currents of opposite signs, where inhibition comes with a short delay. As a result, the time course of the recorded mixed PSC can be complex. Also, the plasticity protocol might affect EPSCs and IPSCs differently, and the estimate of the peak EPSC and its long-term changes might be affected by the overlapping inhibitory currents. To control for possible errors in the estimate of the excitatory peak amplitude, in a set of experiments, we made short applications of the GABA$_A$R antagonist SR95531 (SR) before and after induction, to isolate the excitatory from the inhibitory component of the synaptic response. We assumed that the mixed PSCs are the arithmetic sum of two synaptic functions: an EPSC and a delayed IPSC. For the EPSC we used a function of the form:

$$f(x) = o \, for \, x < xd$$

$$f(x) = B(1 - exp(x - xd2)/tr2)^\wedge 2 * exp(x - xd2)/td2 \, for \, x > xd$$

where x represents time, and A, xd1, td1 and tr1 are the fitting parameters. A is the amplitude, xd1 the delay after stimulus onset, tr1 the rise time constant of the third power exponential rise, and td1 the decay time constant of the EPSC. We used a third-power exponential rise as it best fitted the data compared to a single or quadratic exponential rise.

For the IPSC we used a function of the form:

$$f(x) = o \, for \, x < xd$$

$$f(x) = B(1 - exp(x - xd2)/tr2)^\wedge 2 * exp(x - xd2)/td2 \, for \, x > xd$$

where x is time, and B, xd2, td2 and tr2 are the fitting parameters. B is the amplitude, xd2 the delay after stimulus onset, tr2 the rise time constant of the second power exponential rise, and td2 the decay time constant of the IPSC. For the IPSC a quadratic exponential rise gave a good fit.

The pharmacologically-isolated EPSCs (measured between 2–3 min after SR perfusion) were used to characterise the kinetic parameters of the excitatory responses, and were well described by the fitting function. The fitting EPSC function was then subtracted from the recorded mixed PSC to extract the IPSC component of the response. Since we observed an effect of SR application on the amplitude of EPSCs (presynaptic GABA$_A$Rs are known to modulate release probability *Stell et al., 2007*), the EPSC fitting function was first rescaled to match the rise of the mixed response before subtraction. The resulting inhibitory component of the response was well-fitted by the IPSC function and allowed us to estimate its kinetic parameters. *Appendix 1—figure 1A* shows the average time course of the mixed response normalised to the peak EPSC for both AA and PF, and the extracted excitatory and inhibitory components, during baseline for all cells (n=7). The average delays and decay time parameters estimated were: tau-decay: 5 ± 1 ms for excitatory (n=7), 6.5 ± 1.6 ms for inhibitory (n=6) responses for AA stimulation and 4.4 ± 0.6 ms for excitatory (n=7) and 5 ± 1 ms for inhibitory (n=4) responses for PF stimulation; average delay between inhibitory and excitatory current onset: 1.51 ± 0.15 ms for AA stimulation (n=6), and 1.7 ± 0.3 ms for PF stimulation (n=4).

To measure the excitatory and inhibitory components of the evoked mixed EPSC/IPSC before and after the induction protocol, we first recorded the AA- and PF- mixed PSCs for a short control period and then applied the GABA$_A$ receptor antagonist SR95531 (3 μM) for 3–4 min. SR was then washed off the slice for 15–20 min, the induction protocol was applied, and the synaptic response was monitored for a further 30 min. SR was applied again to monitor the isolated EPSCs at the end of the experiment. For each experiment, the EPSC fitting functions obtained during the first application of SR were used to calculate the amplitude of the two synaptic components during the period preceding induction (10 min), and those obtained from the second SR application at the end of the recordings, to calculate the amplitudes after induction (30 min). *Appendix 1—figure 1B* shows an example of analysis of excitatory and inhibitory components before and after the

induction protocol, with mixed PSCs and corresponding excitatory and inhibitory fitting functions, which shows an increase of the AA-EPSC and a decrease of the PF-EPSC, together with an increase of the IPSC in both pathways. In these experiments, the inhibitory disynaptic component evoked by both AA and PF stimulations had a tendency to increase after the pairing protocol, indicating that excitation and inhibition do not necessarily change in the same direction. However, due to the small sample and the absence of an inhibitory component in some of the experiments, long-term changes in the inhibitory synaptic components were not significantly different from baseline (average change in peak IPSC: 130 ± 50% for the AA pathway, p=0.4 of being bigger than baseline n=5; and 150 ± 50% for the PF pathway, p=0.2 of being bigger than baseline, n=4). When measured at the end of the plasticity experiments, the decay time of the excitatory responses estimated was: tau-decay: 7 ± 2 ms (n=7, not significantly different from baseline, p=0.14) for AA stimulation and 6.0 ± 0.7 ms (n=7, significantly different from baseline, p=0.026) for PF stimulation.

On average, the plasticity of the AA- and PF-EPSC (measured by the usual method of the amplitude of the minimum of the PSC response) in these experiments was not significantly different from that in control experiments (AA pathway: 160 ± 30%, as compared to 131 ± 7% in control, p=0.4, n=7 et n=24 and PF pathway 65 ± 9%, as compared to 65 ± 5% in control, p=1, n=7 et n=25 for SR and control, respectively; the ratio of normalised AA/PF responses: 3 ± 1, p=0.76, n=7 et n=24), showing that the short applications of GABA$_A$R antagonist did not influence the plasticity outcome (*Appendix 1—figure 1*). We thus analysed the change in amplitude of the stimulated pathways using the two separated EPSC/IPSC components obtained by the fitting method before and after the induction protocol to check for differences with respect to the standard method used in all other experiments. *Appendix 1—figure 1* compares, for each cell, the AA- and PF-EPSCs relative amplitudes obtained by direct measure of the mixed PSC minimum with those obtained with the fitting method. The two methods of measurement are equivalent. The delayed onset of the inhibitory component limits its impact on the EPSC amplitude.

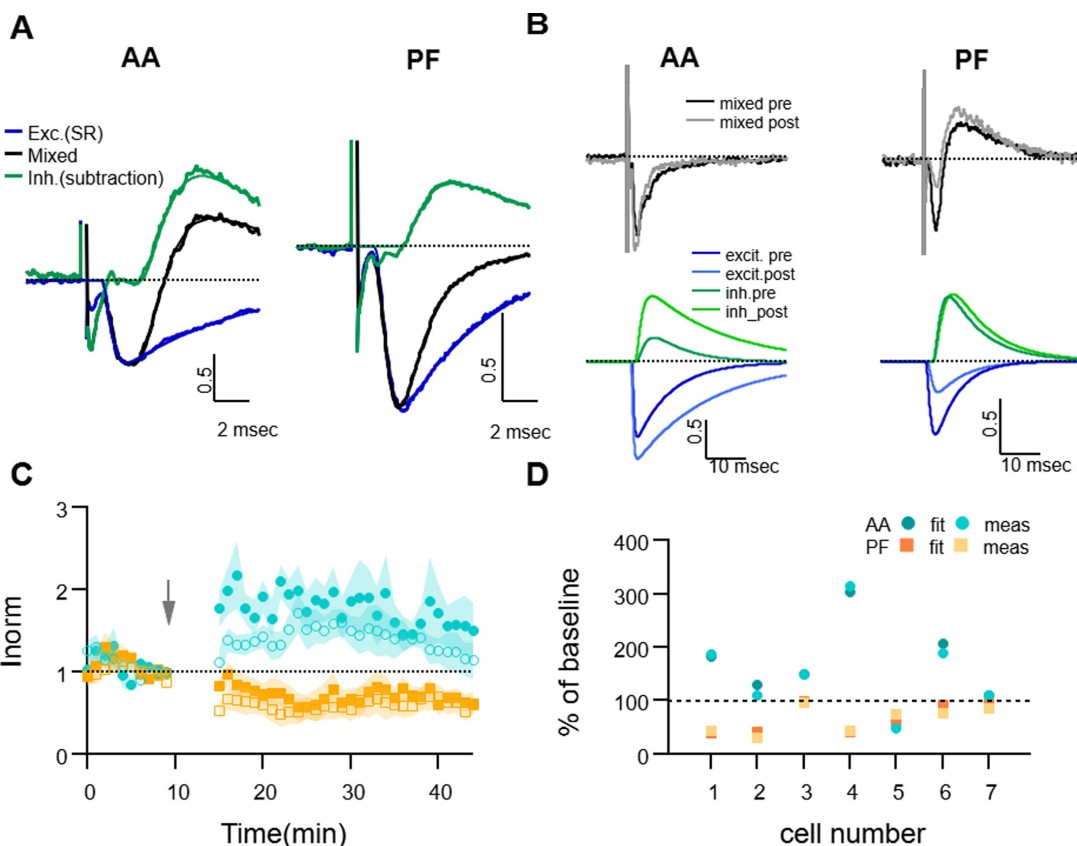

**Appendix 1—figure 1.** Dissecting excitation and inhibition shows that excitatory postsynaptic current (EPSC) amplitude and long term changes are well estimated with the direct measure method. (**A**) Fitting procedure used

*Appendix 1—figure 1 continued*

to extract the excitatory and inhibitory components of the recorded mixed PSCs (EPSC + IPSC). Traces recorded during the baseline period were normalised to the peak excitatory response and then averaged over all cells. The mixed evoked PSC (black thick line) is shown together with the excitatory response recorded during SR application (blue thick line). The inhibitory trace (green thick line) is the difference between the mixed PSC and the fit of the excitatory response. The EPSC and inhibitory postsynaptic current (IPSC) fitting functions as well as the mixed fitting function resulting from the sum of the two are overlaid (thinner lines, same colour), showing good agreement with the data. On average, inhibition was delayed and relatively smaller than excitation for both ascending axon (AA) and parallel fibre (PF) pathways. (**B**) Sample traces for an experiment where the excitatory and inhibitory components were extracted from the mixed PSC for both the baseline and 25–30 min after induction; top: mixed PSCs for AA (left) and PF (right) before and after induction; below are the corresponding excitatory (blue, dark and light for pre and post-induction, respectively) and inhibitory components (green, dark and light for pre and post-induction, respectively). The AA-EPSC increased and the PF-EPSC decreased, while both AA- and PF-IPSCs increased in this example. (**C**) Average time course of the EPSCs relative amplitude (measured with the standard minimum amplitude method) for seven experiments in which gabazine (SR) was applied transiently 20 min before and 30 min after long-term potentiation (LTP) induction. Peak EPSCs amplitudes of the first and second evoked responses in the pair for both AA (blue, closed circles: first response in the pair, open circles: second response) and PF(orange, closed squares: first response in the pair, open squares: second response) stimulation. Short and long-term plastic changes are not different from control experiments. (**D**) The normalised EPSC amplitude 25–30 min after induction is compared for the two analysis methods for each cell, either measured directly as the minimum of the recorded response (meas) or that of the excitatory fitting function (fit). Disynaptic inhibition only slightly affected the measure of the real excitatory peak, both for AA- (blue triangles) and PF-EPSC(orange circles) (AA measured:160 ± 30%, from fit: 160 ± 30% and PF measured: 65 ± 9%, from fit: 70 ± 10%).

