## [Editor Report · eLife assessment]

This study presents **valuable** findings on an unresolved question of cerebellar physiology: Do synapses between Purkinje cells and granule cells, made by the ascending part of the granule cells' axon, have different properties than those made by parallel fibers? The authors conducted patch-clamp recordings on rat cerebellar slices and found a new type of plasticity in the synapses of the ascending part of granule cell axons. The experiments are well-designed with appropriate controls, and the study provides **solid** evidence for the new form of cerebellar synaptic plasticity.

---

## [Referee Report · Reviewer #1 (Public Review)]

In this study, the authors address a fundamental unresolved question in cerebellar physiology: do synapses between granule cells (GCs) and Purkinje cells (PCs) made by the ascending part of the axon (AA) have different synaptic properties to those made by parallel fibers? This is an important question because GCs integrate sensorimotor information from many brain areas with a precise and complex topography.

The authors argue that GCs located close to the PCs essentially contact PC dendrites through the ascending part of their axon. They demonstrate that high-frequency (100 Hz) joint stimulation of distant parallel fibers and local GCs potentiates AA-PC synapses, while parallel fiber-PC synapses are depressed. On the basis of paired pulse ratio analysis, they concluded that evoked plasticity was postsynaptic. When individual pathways are stimulated alone, no LTP is observed. This associative plasticity appears to be sensitive to timing, as stimulation of parallel fibers first results in depression, while stimulation of the AA pathway has no effect. NMDA, mGluR1 and GABAA receptors are involved in this plasticity.

Overall, associative modulation of synaptic transmission is convincing, and the experiments carried out support this conclusion.

One of its weaknesses is that it contradicts the numerous experiments conducted by many groups that have studied plasticity at this connection (e.g. Bouvier et al 2016, Piochon et al 2016, Binda et al, 2016, Schonewille et al 2021). According to the literature, high-frequency stimulation of parallel fibers leads to postsynaptic potentiation under many different experimental conditions (blocked or unblocked inhibition, stimulation protocols, internal solution composition). This discrepancy was not investigated experimentally.

Another weakness is the lack of evidence that AAs have been stimulated. Indeed, without filling the PC with fluorescent dye or biocytin during the experiment, and without reconstructing the anatomical organization, it is difficult to assess whether the stimulating pipette is actually positioned in the GC cluster that potentially contacts the PC with AAs. Although the idea that AAs repeatedly contact the same Purkinje cell has been propagated, to the reviewer's knowledge, no direct demonstration of this hypothesis has yet been published. In fact, what has been demonstrated (Walter et al 2009; Spaeth et al 2022) is that GCs have a higher probability of being connected to nearby PCs, but not necessarily associated with AAs.

---

## [Referee Report · Reviewer #2 (Public Review)]

Summary:

The authors describe a form of synaptic plasticity at synapses from granule cells onto Purkinje cells in the mouse cerebellum, which is specific to synapses from granule cells close to the cell body but not to distal ones. This plasticity is induced by the paired or associative stimulation of the two types of synapses because it is not observed with stimulation of one type of synapse alone. In addition, this form of plasticity is dependent on the order in which the stimuli are presented and is dependent on NMDA receptors, metabotropic glutamate receptors and to some degree on GABAA receptors.

Strengths:

The focus of the authors on the properties of two different synapse-types on cerebellar Purkinje cells is interesting and relevant, given previous results that ascending and parallel fiber synapses might be functionally different and undergo different forms of plasticity (although it hasn't been proven here that the two types of synapses are indeed ascending vs parallel fiber synapses). Nevertheless, the interaction between proximal vs. distal stimulation driven synapse types during plasticity is important for understanding cerebellar function. The demonstration of timing and order-dependent potentiation of only one pathway, and not another, after associative stimulation of both pathways, changes our understanding of potential plasticity mechanisms. In addition, this observation opens up many new questions on underlying intracellular mechanisms as well as on its relevance for cerebellar learning.

Weaknesses:

A concern with this study is that all recordings demonstrate "rundown", a progressive decrease in the amplitude of the EPSC, starting during the baseline period and continuing after the plasticity-induction stimulus. The issues that are causing rundown are not known and may or may not be related to the cellular processes involved in synaptic plasticity. This concern applies in particular to all the experiments where there is a decrease in synaptic strength. However, a key finding of this paper is the associative potentiation of one pathway, which is clearly different from all conditions where there is a decrease in synaptic strength and raises confidence in the authors' conclusions.

In addition, there is some inconsistency with previous results; specifically, that no PF-LTP was induced by PF-alone repeated stimulation.

It remains for future work to identify what these two synapse types, distinguished by the stimulation location, actually are, and where they are on the Purkinje cell dendritic tree. What this specific timing rule is important for is also something that remains to be discovered. Its potential relevance for plasticity and learning will depend on what information these AA vs PF synapses carry, and why their association is meaningful for the circuit and for a behavior. Overall, this study opens up many new questions for the field.

---

## [Referee Report · Reviewer #3 (Public Review)]

Summary:

Granule cells' axons bifurcate to form parallel fibers (PFs) and ascending axons (AAs). While the significance of PFs on cerebellar plasticity is widely acknowledged, the importance of AAs remains unclear. In the current paper, Conti and Auger conducted electrophysiological experiments in rat cerebellar slices and identified a new form of synaptic plasticity in the AA-Purkinje cell (PC) synapses.

Strengths:

The authors applied simultaneous stimulation of AAs and PFs and recorded from PCs and discovered that the strength of AA-PC synapses and PF-PC synapses change in opposite directions: while AA-PC EPSCs increased, PFs-EPSCs decreased. This finding suggests that synaptic responses to AAs and PFs in PCs are jointly regulated, working as an additional mechanism to integrate motor/sensory input. The existence of such plasticity mechanisms may offer new perspectives in studying and modeling cerebellum-dependent behavior. Overall, the experiments are performed well.

Weaknesses:

There are two weaknesses. First, the baseline of electrophysiological recordings is influenced significantly by run-down, limiting the interpretability of the data. Because the amplitude of AA-EPSCs is relatively small, the run-down may have masked some of the changes in EPSCs. However, the authors managed this difficulty using appropriate controls and statistical analysis. Second, while the authors show AA-LTP depends on mGluR, NMDA receptors, and GABA-A receptors, which cell types express these receptors and how they contribute to plasticity is not clarified. Cell-type-specific knockdown of these receptors may clarify this point in future studies.

---

## [Author Response]

The following is the authors’ response to the original reviews.

**Public Reviews:**

**Reviewer #1 (Public Review):**
In this study, the authors address a fundamental unresolved question in cerebellar physiology: do synapses between granule cells (GCs) and Purkinje cells (PCs) made by the ascending part of the axon (AA) have different synaptic properties from those made by parallel fibers? This is an important question, as GCs integrate sensorimotor information from numerous brain areas with a precise and complex topography.Summary:The authors argue that CGs located close to PCs essentially contact PC dendrites via the ascending part of their axons. They demonstrate that joint high-frequency (100 Hz) stimulation of distant parallel fibers and local CGs potentiates AA-PC synapses, while parallel fiber-PC synapses are depressed. On the basis of paired-pulse ratio analysis, they concluded that evoked plasticity was postsynaptic. When individual pathways were stimulated alone, no LRP was observed. This associative plasticity appears to be sensitive to timing, as stimulation of parallel fibers first results in depression, while stimulation of the AA pathway has no effect. NMDA, mGluR1 and GABAA receptors are involved in this plasticity.Strengths:Overall, the associative modulation of synaptic transmission is convincing, and the experiments carried out support this conclusion. However, weaknesses limit the scope of the results.Weaknesses:One of the main weaknesses of this study is the suggestion that high-frequency parallel-fiber stimulation cannot induce long term potentiation unless combined with AA stimulation. Although we acknowledge that the stimulation and recording conditions were different from those of other studies, according to the literature (e.g. Bouvier et al 2016, Piochon et al 2016, Binda et al, 2016, Schonewille et al 2021 and others), high-frequency stimulation of parallel fibers leads to long-term postsynaptic potentiation under many different experimental conditions (blocked or unblocked inhibition, stimulation protocols, internal solution composition). Furthermore, in vivo experiments have confirmed that high-frequency parallel fibers are likely to induce long-term potentiation (Jorntell and Ekerot, 2002; Wang et al, 2009).This article provides further evidence that long-term plasticity (LTP and LTD) at this connection is a complex and subtle mechanism underpinned by many different transduction pathways. It would therefore have been interesting to test different protocols or conditions to explain the discrepancies observed in this dataset.

Even though this is not the main result of this study, we acknowledge that the control experiments done on PF stimulation add a puzzling result to an already contradictory literature. High frequency parallel fibre stimulation (in isolation) has been shown to induce long term potentiation in vitro, but not always, and most importantly, this has been shown in vivo. This was the reason for choosing that particular stimulation protocol. Examination of in vitro studies, however, show that the results are variable and even contradictory. Most were done in the presence of GABAA receptor antagonists, including the SK channel blocker Bicuculline, whereas in the study by Binda (2016), LTP was blocked by GABAA receptor inhibition. In some studies also, LTP was under the control of NMDAR activation only, whereas in Binda (2016), it was under the control of mGluR activation. Moreover, most experiments were done in mice, whereas our study was done in rats. Our results reveal multiple mechanisms working together to produce plasticity, which are highly sensitive to in vitro conditions. We designed our experiments to be close to the physiological conditions, with inhibition preserved and a physiological chloride gradient. It is likely that experimental differences have given rise to the variability of the results and our inability to reproduce PF-LTP, but it was not the aim of this study to dissect the subtleties of the different experimental protocols and models.

We have modified the Discussion to cover that point fully.

Another important weakness is the lack of evidence that the AAs were stimulated. Indeed, without filling the PC with fluorescent dye or biocytin during the experiment, and without reconstructing the anatomical organization, it is difficult to assess whether the stimulating pipette is positioned in the GC cluster that is potentially in contact with the PC with the AAs. According to EM microscopy, AAs account for 3% of the total number of synapses in a PC, which could represent a significant number of synapses. Although the idea that AAs repeatedly contact the same Purkinje cell has been propagated, to the best of the review author's knowledge, no direct demonstration of this hypothesis has yet been published. In fact, what has been demonstrated (Walter et al 2009; Spaeth et al 2022) is that GCs have a higher probability of being connected to nearby PCs, but are not necessarily associated with AAs.

We fully agree with the reviewer that we have not identified morphologically ascending axon synapses, and we stress this fact both in the first paragraph of the Results section, and again at the beginning of Discussion. Our point is mainly topographical, given the well documented geometrical organisation of the cerebellar cortex. Strictly speaking, inputs are local (including AAs) or distal (PFs). Similarly, the studies by Isope and Barbour (2002) and Walter et al. (2009), just like Sims and Hartell (2005 and 2006), have coined the term ‘ascending axon’ when drawing conclusions about locally stimulated inputs. Moreover, our results do not rely on or assume multiple contacts, stronger connections, or higher probability of connections between ascending axons and Purkinje cells. Our results only demonstrate a different plasticity outcome for the two types of inputs. Therefore, our manuscript could be rephrased with the terms ‘local’ and ‘distal’ granule cell inputs, but this would have no more implication for the results or the computation performed in Purkinje cells. However, in our experience, these terms are more confusing, and consistent with the literature, we do not wish to make this modification. However, we have modified the abstract of the manuscript to clarify this point.

**Reviewer #2 (Public Review):**
Summary:The authors describe a form of synaptic plasticity at synapses from granule cells onto Purkinje cells in the mouse cerebellum, which is specific to synapses proximal to the cell body but not to distal ones. This plasticity is induced by the paired or associative stimulation of the two types of synapses because it is not observed with stimulation of one type of synapse alone. In addition, this form of plasticity is dependent on the order in which the stimuli are presented, and is dependent on NMDA receptors, metabotropic glutamate receptors and to some degree on GABAA receptors. However, under all experimental conditions described, there is a progressive weakening or run-down of synaptic strength. Therefore, plasticity is not relative to a stable baseline, but relative to a process of continuous decline that occurs whether or not there is any plasticity-inducing stimulus.

As highlighted by the reviewer, we observed a postsynaptic rundown of the EPSC amplitude for both input pathways. Rundown could be mistaken for a depression of synaptic currents, not for a potentiation, and the progressive decrease of the EPSC amplitude during the course of an experiment leads to an underestimate of the absolute potentiation. We have taken the view to provide a strong set of control data rather than selecting experiments based on subjective criteria or applying a cosmetic compensation procedure. We have conducted control experiments with no induction (n = 17), which give a good indication of the speed and amplitude of the rundown. Comparison shows a highly significant potentiation of the ascending axon EPSC. Depression of the parallel fibre EPSC, on the other hand, was not significantly different from rundown, and we have not spoken of parallel fibre long term depression. The data show thus very clearly that ascending axon and parallel fibre synapses behave differently following the costimulation protocol.

Strengths:The focus of the authors on the properties of two different synapse-types on cerebellar Purkinje cells is interesting and relevant, given previous results that ascending and parallel fiber synapses might be functionally different and undergo different forms of plasticity. In addition, the interaction between these two synapse types during plasticity is important for understanding cerebellar function. The demonstration of timing and order-dependent potentiation of only one pathway, and not another, after associative stimulation of both pathways, changes our understanding of potential plasticity mechanisms. In addition, this observation opens up many new questions on underlying intracellular mechanisms as well as on its relevance for cerebellar learning and adaptation.Weaknesses and suggested improvements:A concern with this study is that all recordings demonstrate "rundown", a progressive decrease in the amplitude of the EPSC, starting during the baseline period and continuing after the plasticity-induction stimulus. In the absence of a stable baseline, it is hard to know what changes in strength actually occur at any set of synapses. Moreover, the issues that are causing rundown are not known and may or may not be related to the cellular processes involved in synaptic plasticity. This concern applies in particular to all the experiments where there is a decrease in synaptic strength.

We have provided an answer to that point directly below the summary paragraph. We will just add here that if the phenomenon causing rundown was involved in plasticity, it should affect plasticity of both inputs, which was not the case, clearly distinguishing the ascending axon and parallel fibre inputs.

The authors should consider changes in the shape of the EPSC after plasticity induction, as in Fig 1 (orange trace) as this could change the interpretation.

Figure 1 shows an average response composed of evoked excitatory and inhibitory synaptic currents. The third section of Supplementary material (supplementary figure 3) shows that this complex shape is given by an EPSC followed by a delayed disynaptic IPSC. We would like to point out that while separating EPSC from IPSC might appear difficult from average traces due to the averaged jitter in the onset of the synaptic currents, boundaries are much clearer when analysing individual traces. In the same section we discuss the results of experiments in which transient applications of SR 95531 before and after the induction protocol allowed us to measure the EPSC, while maintaining the same experimental conditions during induction. Analysis of the kinetics of the EPSCs during SR application at the beginning and end of experiments, showed that there is no change in the time to peak of both AA and PF response. The decay time of AA- and PF-EPSCs are slightly longer at the end of the experiment, even if the difference is not significant for AA inputs. This analysis has been added to the Supplementary material. Our analysis, that uses as template the EPSCs kinetics measured at the beginning and at the end of the experiments, takes directly into account these changes. The results show clearly that the presence of disynaptic inhibition doesn’t significantly affect the measure of the peak EPSC after the induction protocol nor the estimate of plasticity.

In addition, the inconsistency with previous results is surprising and is not explained; specifically, that no PF-LTP was induced by PF-alone repeated stimulation.

In our experimental conditions, PF-LTP was not induced when stimulating PF only, the condition that reproduces experiments in the literature. As discussed in our response to reviewer 1, a close look at the literature, however, reveals variabilities and contradictions behind seemingly similar results. They reveal intricate mechanisms working together to produce plasticity, which are sensitive to in vitro conditions. We designed our experiments to be close to physiological conditions, with inhibition preserved and a physiological chloride gradient. It is likely that experimental differences have given rise to the variability of the results and our inability to observe PF-LTP. We have modified the Discussion section to cover that point thoroughly in the context of past results.

The authors test the role of NMDARs, GABAARs and mGluRs in the phenotype they describe. The data suggest that the form of plasticity described here is dependent on any one of the three receptors. However, the location of these receptors varies between the Purkinje cells, granule cells and interneurons. The authors do not describe a convincing hypothetical model in which this dependence can be explained. They suggest that there is crosstalk between AA and PF synapses via endocannabinoids downstream of mGluR or NO downstream of NMDARs. However, it is not clear how this could lead to the long-term potentiation that they describe. Also, there is no long-lasting change in paired-pulse ratio, suggesting an absence of changes in presynaptic release.

We suggest in the result section that the transient change in paired pulse ratio (PPR) is linked to a transient presynaptic effect, but there was no significant long term change of the PPR, suggesting that the long term effects observed are linked to postsynaptic changes. We now stress this point in the Results and Discussion sections.

Concerning the involvement of multiple molecular pathways, investigators often tested for the involvement of NMDAR or mGluRs in cerebellar plasticity, rarely both. Here we showed that both pathways are involved. The conjunctive requirement for NMDAR and mGluR activation could easily be explained based on the dependence of cerebellar LTP and LTD on the concentrations of both NO and postsynaptic calcium (Coesman et al., 2004; Safo and Regehr, 2005; Bouvier et al., 2016; Piochon et al., 2016).

We also observed an effect of GABAergic inhibition. GABAergic inhibition was elegantly shown by Binda (2016) to regulate calcium entry together with mGluRs, and control plasticity induction. A similar mechanism could contribute to our results, although inhibition might have additional effects. We have modified the Discussion of the manuscript to clarify the pathways involved in plasticity and added a diagram to highlight the links between the different molecular pathways, potential cross talk mechanisms, and the location of receptors.

Is the synapse that undergoes plasticity correctly identified? In this study, since GABAergic inhibition is not blocked for most experiments, PF stimulation can result in both a direct EPSC onto the Purkinje cell and a disynaptic feedforward IPSC. The authors do address this issue with Supplementary Fig 3, where the impact of the IPSC on the EPSC within the EPSC/IPSC sequence is calculated. However, a change in waveform would complicate this analysis. An experiment with pharmacological blockade will make the interpretation more robust. The observed dependence of the plasticity on GABAA receptors is an added point in favor of the suggested additional experiments.

We did consider that due to long recording times there might be kinetic changes, and that’s the reason why the experiments of Supplementary figure 3 were done with pharmacological blockade of GABAAR with SR, both before and again after LTP induction. The estimate of the amplitude of the EPSC is based on the actual kinetics of the response at both times.

A primary hypothesis of this study is that proximal, or AA, and distal, or PF, synapses are different and that their association is specifically what drives plasticity. The alternative hypothesis is that the two synapse-types are the same. Therefore, a good control for pairing AA with PF would be to pair AA with AA and PF with PF, thereby demonstrating that pairing with each other is different from pairing with self.

Pairing AA with AA would be difficult because stimulation of AA can only be made from a narrow band below the PC and we would likely end up stimulating overlapping sets of synapses. However, Figure 5 shows the effect of stimulating PF and PF, while also mimicking the sparse and dense configuration of the control experiment. It shows that sparse PF do not behave like AA. Sims and Hartell (2006) also made an experiment with sparse PF inputs and observed clear differences between sparse local (AA) and sparse distal (PF) synapses.

It is hypothesized that the association of a PF input with an AA input is similar to the association of a PF input with a CF input. However, the two are very different in terms of cellular location, with the CF input being in a position to directly interact with PF-driven inputs. Therefore, there are two major issues with this hypothesis: (1) how can subthreshold activity at one set of synapses affect another located hundreds of micrometers away on the same dendritic tree? (2) There is evidence that the CF encodes teaching/error or reward information, which is functionally meaningful as a driver of plasticity at PF synapses. The AA synapse on one set of Purkinje cells is carrying exactly the same information as the PF synapses on another set of Purkinje cells further up and down the parallel fiber beam. It is suggested that the two inputs carry sensory vs. motor information, which is why this form of plasticity was tested. However, the granule cells that lead to both the AA and PF synapses are receiving the same modalities of mossy fiber information. Therefore, one needs to presuppose different populations of granule cells for sensory and motor inputs or receptive field and contextual information. As a consequence, which granule cells lead to AA synapses and which to PF synapses will change depending on which Purkinje cell you're recording from. And that's inconsistent with there being a timing dependence of AA-PF pairing in only one direction. Overall, it would be helpful to discuss the functional implications of this form of plasticity.

We do not hypothesise that association of the AA and PF inputs is similar to the association of PF and climbing fibre inputs. We compare them because it is the other known configuration triggering associative plasticity in Purkinje cells. It is indeed interesting to observe that even if the inputs are very small compared to the powerful climbing fibre input, they can be effective at inducing plasticity. Physiologically, the climbing fibre signal has been linked to error and reward signals, but reward signals are also encoded by granule cell inputs (Wagner et al., 2017). We have modified the discussion to make sure that we do not suggest equivalence with CF induced LTD.

Moreover, we fully agree that AA and PF synapses made up by a given granule cell carry the same information, and cannot encode sensory and motor information at the same time. AA synapses from a local granule cell deliver information about the local receptive field, but PF synapses from the same granule cell will deliver contextual information about that receptive field to distant Purkinje cells. In the context of sensorimotor learning, movement is learnt with respect to a global context, not in isolation, therefore learning a particular association must be relevant. The associative plasticity we describe here could help explain this functional association. We have clarified the discussion.

**Reviewer #3 (Public Review):**
Granule cells' axons bifurcate to form parallel fibers (PFs) and ascending axons (AAs). While the significance of PFs on cerebellar plasticity is widely acknowledged, the importance of AAs remains unclear. In the current paper, Conti and Auger conducted electrophysiological experiments in rat cerebellar slices and identified a new form of synaptic plasticity in the AA-Purkinje cell (PC) synapses. Upon simultaneous stimulation of AAs and PFs, AA-PC EPSCs increased, while PFs-EPSCs decreased. This suggests that synaptic responses to AAs and PFs in PCs are jointly regulated, working as an additional mechanism to integrate motor/sensory input. This finding may offer new perspectives in studying and modeling cerebellum-dependent behavior. Overall, the experiments are performed well. However, there are two weaknesses. First, the baseline of electrophysiological recordings is influenced significantly by run-down, making it difficult to interpret the data quantitatively. The amplitude of AA-EPSCs is relatively small and the run-down may mask the change. The authors should carefully reexamine the data with appropriate controls and statistics. Second, while the authors show AA-LTP depends on mGluR, NMDA receptors, and GABA-A receptors, which cell types express these receptors and how they contribute to plasticity is not clarified. The recommended experiments may help to improve the quality of the manuscript.

As highlighted by the reviewer and developed above in response to reviewer 2, we observed a postsynaptic rundown of the EPSC amplitude. Rundown could be mistaken for a depression of synaptic currents, not for a potentiation. Moreover, we have conducted control experiments with no induction (n = 17), which give a good indication of the speed and amplitude of the rundown, and provide a baseline. Comparison shows a highly significant potentiation of the ascending axon EPSC, relative to baseline and relative to these control experiments. Depression of the parallel fibre EPSC on the other hand was not significantly different from rundown. For that reason we have not spoken of parallel fibre long term depression. The data, however, show that ascending axon and parallel fibre synapses behave very differently following the costimulation protocol.

We have discussed above in our response to reviewer 2 the potential involvement of mGluRs, NMDARs and GABAARs. We have clarified the discussion of the pathways involved in plasticity and added a diagram to highlight the links between the different molecular pathways, potential cross talk mechanisms, and the location of receptors.

**Recommendations for the authors:**

**Reviewer #1 (Recommendations For The Authors):**
- If Chloride concentration cannot be modified, recordings should be performed at the Chloride reversal potential to avoid strong bias in amplitude measurements (e.g. in Figures 3 and 5 outward current was observed while not visible in Figures 1 and 4).

The balance between excitation and inhibition dictates whether there is a visible outward component, and this varies with the connections tested. Careful control experiments with SR application presented in supplementary figure 3 show that the delay of the IPSC does not significantly affect measurement of the peak amplitude of the EPSC. The reversal potential for Clin our study (-85 mV), chosen to reproduce the physiological gradient in Purkinje cells, is too low to record from Purkinje cells at this potential in good conditions as it activates the hyperpolarisation activated cation current Ih, generating huge inward currents.

- It is not clear whether, during the current clamp, the potential was maintained at -65 mV throughout the induction protocol.

The potential was set and maintained around -65mV during the induction protocol. The method section has been amended to specify that point.

- Experiments using GABAB or endocannabinoid antagonists would have been interesting to assess the role of presynaptic plasticity occluding postsynaptic plasticity.

We are not sure why the reviewer suggested these particular experiments to test for the role of presynaptic plasticity. GABAB and endocannabinoid receptor activation both have presynaptic effects at granule cell to Purkinje cell synapses. They decrease release probability, and as a result increase the paired pulse ratio (Dittman and Regehr, 1997; Safo and Regehr, 2005). Here we only observed a transient decrease of the paired pulse ratio. Additionally, presynaptic endocannabinoid receptor activation, linked to postsynaptic mGluR1 activation and release of endocannabinoids, was shown to be required for induction of postsynaptic PF-LTD (Safo and Regehr, 2005). This effect required climbing fibre stimulation and mGluR activation. Here we show that mGluR1 inhibition did not inhibit the PF depression nor affect the transient change in PPR. Therefore there is no indication that activation of these receptors could induce a pre-synaptic depression occluding postsynaptic plasticity.

- To give credit to this new plasticity in contradiction with many previous studies, induction pathways should be addressed more deeply.

As developed earlier in response to the public review, this study does not contradict previous studies, expect maybe that by Binda et al., (2016), conducted on mice. From our point of view, our study in fact reconciles past results which have alternatively involved the mGluR or NMDAR pathways, whereas the molecular downstream pathways they recruit can easily cooperate. We aim to describe a new phenomenon and we cannot cover the mechanistic dissection which has been performed to date on plasticity in the cerebellar cortex.

- The quality of the figures could be enhanced by modifying the dashed line.

We have made the dashed line more discrete.

**Reviewer #2 (Recommendations For The Authors):**
- Is there cross-talk between the two synaptic pathways?

In order to explain the associative nature of AA-LTP we suggest that a signal is generated at the AA input during the induction protocol only when the PF input is also stimulated, i.e. a form of cross-talk takes place between the two synaptic territories. We have not tested for cross-talk during control conditions but we discuss the fact that given the size of the Purkinje cell dendritic tree, the size of the inputs and their geometrical configuration, it is highly unlikely. We discuss possible cross-talk mechanisms.

- Clarification question: "While the peak amplitude of the first response in the pair of stimulations showed a progressive decline, the peak amplitude of the second response of both AA and PF underwent either LTP or LTD respectively..." Does this mean that all LTP/LTD figures show the amplitude of the second EPSC in the paired pulse stimulation, and that the first EPSC has a different response? If so, this should be mentioned in the Methods section and implications discussed.

All figures show both the amplitude of the first and second EPSCs in the pair of stimulations. In Figure 1A, 3A, 4A and 5B the paired stimulation protocol is depicted with colours and symbols used in the associated graphs, with closed symbols for the first and open symbols for the second EPSC. Figure legends have been amended to clarify this point. The average values given in the Results section and figure legends relate to the first EPSC only for clarity. As can be seen from the figures, long term plasticity affected the first and second EPSC in a very similar manner. However, individual symbols show that during a transient period, the first and second EPSCs are differentially affected by the induction protocol, resulting in a transient change of the PPR.

Minor suggestions:- It would be helpful to have a reference for the statement that 1-2% of stimulated fibers come from nearby GCs when stimulation is distal.

We have modified the text to explain our calculation based on the data of Pichitpornchai et al., 1994. P4 result section.

- Does the shading over the plasticity time course traces come from the standard error of the mean?

Shading over the plasticity time course plots shows the standard error of the mean. This is now clearly stated in figure legends.

**Reviewer #3 (Recommendations For The Authors):**
Major points:(1) Whether the plasticity between AAs and PCs is regulated by the post-synaptic or pre-synaptic mechanisms should be addressed or discussed. Based on the results of PPR (mostly unchanged after induction), the post-synaptic mechanism may be more significant. Supplemental Figure 2C shows a trend toward a positive correlation between AALTP and the number of spikes, suggesting intracellular calcium levels in the post-synaptic Purkinje cells may be important. Whether this is true or not can be directly tested by the addition of BAPTA in the recording pipettes.

The absence of a long lasting effect on the paired pulse ratio (PPR) indicates that postsynaptic mechanisms are involved in long term changes. This is in line with the dependence of plasticity induced with similar protocols on the concentrations of NO and postsynaptic calcium, both affecting postsynaptic targets, as developed in our response to reviewer 2. BAPTA interferes with calcium and mGluR signalling, and could be used to further confirm the involvement of a postsynaptic mechanism, however, we did not wish to pursue further the dissection of the signalling cascade. We have modified the Results and Discussion sections to include a discussion of pre and postsynaptic mechanisms.

(2) Most results from the plasticity experiments are shown as average/sem and do not include individual data, making ithard to appreciate the magnitude of the changes. The authors could show the individual data at some time points (e.g. 5 min before and 30 min after induction), plot bar-graphs (Figure 2C with individual data), or boxplots to compare different conditions and perform statistics.

Individual data points are now visible for plasticity induction in Figure 2C and Supplementary Figure 2 for a number of conditions. Statistics have been performed as detailed in the text and legend of Fig 2.

(3) In addressing point #2, it is strongly recommended that the authors include the values for controls without inductionbecause AA/PF-EPSCs undergo significant run-down. In most experiments, the authors compare the magnitude of plasticity with baseline changes in Supplemental Figure 1. This should not be appropriate for some experiments, such as Figures 3 & 4, where pharmacological treatments are performed. The authors should carefully consider including the appropriate controls from baseline recording to rule out significant confound by the run-down.

We agree that control experiments without stimulation (no Stim) are only appropriate controls for the initial synchronous stimulation and AA and PF only experiments (Fig 1). All the other experiments were compared to the synchronous stimulation experiments, not to control No Stim. The synchronous stimulation protocol is strictly the same as that applied in experiments with pharmacological treatments and the appropriate control to test whether treatments affected plasticity. This is now systematically specified in the Results section.

(4) The authors recorded mixed EPSC/IPSCs and used a fitting approach to extract EPSCs. Applying AMPA-receptor blockers to check that extracted IPSCs are correctly predicted may solidify the reliability of the approach. An additional concern is that this approach can only be used if the waveform of EPSC/IPSC does not change with plasticity. The authors should compare the waveforms between conditions to address this point.

Fits were not used to extract EPSCs. EPSCs were isolated by blocking IPSCs with SR95531, and the IPSCs were then extracted by subtraction from the mixed EPSC/IPSC. Fits were then done of the isolated EPSC and the extracted IPSC. This procedure was applied both at the start of the experiment and at the end to avoid changes in kinetics that would influence measurements. A section of supplementary material is devoted to this analysis. Isolating IPSCs using AMPAR blockers is not possible as IPSCs are disynaptic. AMPAR blockers would fully suppress inhibition.

(5) While the AA-LTP depends on NMDA-Rs, which cell type is responsible is not clear. Recording NMDA components in AA/PF-EPSCs should be informative in addressing this point. Cesana et al suggested that AA induces significant activation of NMDA-Rs in Golgi cells (PMID: 23884948). Whether AA stimuli could significantly evoke NMDA current in the experimental condition used in this paper could provide essential information.

The granule cell to Purkinje cell EPSCs are devoid of an NMDAR component (Llano et al., 1991), and there is no postsynaptic NMDARs at granule cell to PC synapses, but a proportion of presynaptic boutons show the presence of NMDARs (Bidoret et al, 2009). This is now stated clearly on p8. Presynaptic NMDAR have been involved in LTP and LTD of parallel fibre synapses (Casado et al., 2002; Bouvier et al., 2016; Schonewille et al., 2021), and linked to the activation of NOS in granule cell axons. However, we do not know whether presynaptic NMDARs are also present at AA synapses. NMDAR and NOS are also expressed by molecular layer interneurons, and have sometimes been involved in LTD induction (Kono et al., 2019), although this is disputed. In the paper by Cesana (2013), white matter stimulation activated mossy fibre inputs to granule cells, and as a consequence, granule cell to Golgi cell disynaptic EPSCs. The authors identified AA synapses on the basolateral dendrites of Golgi cells, and showed NMDAR activation associated with the mossy fibre to granule cell EPSC. Granule cell to Golgi cell synapses were shown to activate both postsynaptic AMPA and NMDA receptors (Dieudonné, 1999). But to our knowledge, Golgi cells do not express NOS. Therefore it is unlikely that activation of NMDARs in Golgi cells is linked to synaptic plasticity in Purkinje cells.

(6) Pharmacological experiments in Figure 3 show that AA-LTP is dependent on mGluR. The authors mentioned that it could be explained by the presence and absence of mGluRs in PFs and AAs, respectively. This is an important and reasonable possibility and should be tested. The authors could simply check whether slow EPSCs can be recorded by the AA activation.

Activation of the mGluR slow EPSC by AA stimulation would reveal the presence of mGluRs at AA inputs. We know, however, that sparse PF stimulation does not activate the mGluR slow EPSC nor endocannabinoid release unless glutamate transporters are blocked (Marcaggi and Attwell., 2005). This is thought to reflect insufficient glutamate buildup in the sparse configuration to activate mGluR1s. AA inputs are sparsely distributed and are not expected to activate the slow EPSC either, and this is confirmed by our own experiments (CA personal communication). However, mGluR1 mediated Ca2+ release from stores shows a higher sensitivity to glutamate than the slow EPSC (Canepari and Ogden, 2006) and might take place with sparse inputs, but Ca2+ signals have not been investigated in this configuration. Therefore the absence of the slow EPSC is not sufficient proof that mGluR1s are not activated and not present at AA synapses. This is now further discussed p12.

Minor points:(1) The authors should describe how they adjusted the stimulation strength for both AAs and PFs.

Adjustment of the stimulation intensity is now described in the Methods section.

(2) A rationale explaining why the authors chose the current induction protocol (synchronous stimulation of both inputs) should be included. This will help the readers to understand the background of the study.

Papers by Sims and Hartell (2005, 2006) and experimental evidence indicated that AA and PF inputs may have different properties, and as a result may play different roles. Moreover, based on the morphology of the cerebellar granule cell and Purkinje cell, AA and PF inputs can carry different information to a given Purkinje cell. We reasoned that co-presentation of the inputs might represent an important piece of information for the circuit, signalling functional association, and lead to plasticity, as seen for motor command and sensory feedback in cerebellar-like structures, or for PF and climbing fibre. We have tried to convey that rational in the abstract and introduction.

(3) Supplemental Figure 2B: the x-axis may be labeled incorrectly, Is the x-axis of the top graph for PF PF-EPSC? Thex-axis for the bottom graphs should be the summation of AA- and PF-EPSCs.

This has been corrected.

(4) "mglur1" on page 10 should be mGluR1.

This has been corrected.